# Experimental factors that impact Ca$_V$1.2 channel pharmacology—Effects of recording temperature, charge carrier, and quantification of drug effects on the step and ramp currents elicited by the "step-step-ramp" voltage protocol

**Ming Ren**[1☯]**, Aaron L. Randolph**[1,2☯]**, Claudia Alvarez-Baron**[1]**, Donglin Guo**[3]**, Phu N. Tran**[1,4]**, Nicolas Thiebaud**[1,5]**, Jiansong Sheng**[1,6]**, Jun Zhao**[1]**, Wendy W. Wu**[1]*

**1** Division of Applied Regulatory Science, Office of Clinical Pharmacology, Center for Drug Evaluation and Research, US Food and Drug Administration, Silver Spring, Maryland, United States of America, **2** Nanion Technologies Inc, Livingston, New Jersey, United States of America, **3** Division of Cardiology and Nephrology, Office of Cardiology, Hematology, Endocrinology and Nephrology, Office of New Drugs, Center for Drug Evaluation and Research, US Food and Drug Administration, Silver Spring, Maryland, United States of America, **4** Division of Immunology and Hematology Devices, Center for Devices and Radiological Health, US Food and Drug Administration. Silver Spring, Maryland, United States of America, **5** Vertex Pharmaceuticals (Europe) Ltd, Abingdon, Oxfordshire, United Kingdom, **6** CiPA Lab, Gaithersburg, Maryland, United States of America

☯ These authors contributed equally to this work.
* wendy.wu@fda.hhs.gov

## Abstract

### Background and purpose

Ca$_V$1.2 channels contribute to action potential upstroke in pacemaker cells, plateau potential in working myocytes, and initiate excitation-contraction coupling. Understanding drug action on Ca$_V$1.2 channels may inform potential impact on cardiac function. However, literature shows large degrees of variability between Ca$_V$1.2 pharmacology generated by different laboratories, casting doubt regarding the utility of these data to predict or interpret clinical outcomes. This study examined experimental factors that may impact Ca$_V$1.2 pharmacology.

### Experimental approach

Whole cell recordings were made on Ca$_V$1.2 overexpression cells. Current was evoked using a "step-step-ramp" waveform that elicited a step and a ramp current. Experimental factors examined were: 1) near physiological vs. room temperature for recording, 2) drug inhibition of the step vs. the ramp current, and 3) Ca$^{2+}$ vs. Ba$^{2+}$ as the charge carrier. Eight drugs were studied.

### Key results

Ca$_V$1.2 current exhibited prominent rundown, exquisite temperature sensitivity, and required a high degree of series resistance compensation to optimize voltage control. Temperature-

**Data Availability Statement:** To support data transparency, the original electrophysiology records, detailed cell culture procedure, and supplemental materials for the present study are available for download at: https://osf.io/g3msb/.

**Funding:** This study was supported by the operating budget of the Division of Applied Regulatory Science at the United States Food and Drug Administration. The funder had no role in study design, data collection and analysis, decision to publish, or preparation of the manuscript.

**Competing interests:** The authors have declared that no competing interests exist.

dependent effects were examined for verapamil and methadone. Verapamil's block potency shifted by up to 4X between room to near physiological temperature. Methadone exhibited facilitatory and inhibitory effects at near physiological temperature, and only inhibitory effect at room temperature. Most drugs inhibited the ramp current more potently than the step current—a preference enhanced when $Ba^{2+}$ was the charge carrier. The slopes of the concentration-inhibition relationships for many drugs were shallow, temperature-dependent, and differed between the step and the ramp current.

## Conclusions and implications

All experimental factors examined affected Ca$_V$1.2 pharmacology. In addition, whole cell Ca$_V$1.2 current characteristics—rundown, temperature sensitivity, and impact of series resistance—are also factors that can impact pharmacology. Drug effects on Ca$_V$1.2 channels appear more complex than simple pore block mechanism. Normalizing laboratory-specific approaches is key to improve inter-laboratory data reproducibility. Releasing original electrophysiology records is essential to promote transparency and enable the independent evaluation of data quality.

## Introduction

Ca$_V$1.2 channels in the heart mediate L-type $Ca^{2+}$ current that contributes to $Ca^{2+}$-dependent action potentials (APs) in the pacemaker cells of the sinoatrial and atrioventricular nodes, the plateau phase of the AP in the working myocytes, and triggers cardiac excitation-contraction coupling [1]. Drugs that reduce Ca$_V$1.2 channel activity slow heart rate, shorten AP duration in atrial and ventricular myocytes, and decrease contractile force. Ca$_V$1.2 channel agonists have not been used in humans. Nonetheless, studies have shown that pharmacologically enhancing L-type $Ca^{2+}$ current can produce delayed repolarization and ventricular arrhythmias [2, 3]. Therefore, understanding drug effects on Ca$_V$1.2 channels may provide insights regarding a drug's impact on cardiac function. Indeed, a survey of nonclinical safety assessment of proarrhythmia risk used by the pharmaceutical industry found that during early drug discovery, patch clamp characterization of a drug candidate's effect on Ca$_V$1.2 channels is routinely performed, with frequency second only to the hERG assay [4].

Despite of the potential of Ca$_V$1.2 data to inform drug effect in the clinical setting, leveraging *in vitro* patch clamp results has been challenging for drug regulators. Because of the high inter-laboratory data variability reported in the literature, a major concern is that conclusions based on these data are laboratory-dependent. Inter-laboratory data variability is often ascribed to the use of different voltage protocols and stimulation frequencies. However, results from two recent publications suggest that additional factors may be involved [5, 6]. In these studies, Ca$_V$1.2 current was recorded using overexpression cell lines, and evoked from the same holding potential, at the same frequency (0.1 Hz), using either a ventricular AP waveform [5] or a ventricular AP-like "step-step-ramp" waveform [6]. Nonetheless, surprisingly different inhibitory potencies were obtained for the same drugs, with difference up to 1743X reported (Table 1). These results underscore the need to understand the conduct and design of Ca$_V$1.2 experiments—how each experimental factor may translate into differences in drug effect, especially if these *in vitro* results are to be used in decision-making regarding risk prediction or mechanistic interpretation of clinical outcomes.

Table 1. $IC_{50}$ differences for $Ca_V1.2$ channel block reported by Crumb et al. 2016 [5] and Li et al. 2018 [6].

| | Crumb et al. 2016 [5] | | Li et al. 2018 [6] | | Ratio (max-to-min) | Log P |
|---|---|---|---|---|---|---|
| | $IC_{50}$ (μM) | $n_H$ | $IC_{50}$ (μM) | $n_H$ | | |
| **Bepridil** | 2.8 | 0.6 | 638 | 4.6 | 228 | 5.49 |
| **Chlorpromazine** | 8.2 | 0.8 | 6.35 | 2 | 1.3 | 4.54 |
| **Diltiazem** | 0.1 | 0.7 | 31.6 | 1.2 | 316 | 2.73 |
| **Ondansetron** | 22.6 | 0.8 | 9310 | 0.2 | 412 | 2.35 |
| **Terfenadine** | 0.7 | 0.7 | 1220 | 5.2 | 1743 | 6.48 |
| **Verapamil** | 0.2 | 1.1 | 11.2 | 0.8 | 56 | 5.04 |

Comparison of the abovementioned publications revealed several differences in the $Ca_V1.2$ experimental conduct and design. One study used a manual patch clamp system, recorded cells at near physiological temperature (PT) using $Ba^{2+}$ as the charge carrier, and quantified drug effect on the inward current associated with the repolarizing phase of the AP [5]. The other used an automated patch clamp system, recorded cells at ambient temperature using $Ca^{2+}$ as the charge carrier, and quantified drug effect on the inward current triggered by the initial voltage step [6]. Recording temperature [7, 8] and charge carrier [9–12] are known to affect block potencies for some drugs on $Ca_V1.2$ channels in overexpression cells and L-type channels in native myocytes. In addition, measuring drug effects on $Ca_V1.2$ current evoked at different time points following the initial depolarization and associated with different membrane voltages (i.e., current resulting from different channel state, due to activation [6] or reactivation following recovery from inactivation [5]), as was done in these two studies is also a likely source of data variability, given that many drugs are known to block $Ca_V1.2$ and L-type $Ca^{2+}$ channels in a state-dependent manner [13–17].

Using manual whole cell patch clamp method to record cells stably expressing $hCa_V1.2α$, $β_2$, and $α_2δ_1$ subunits, the present study was conducted to examine $Ca_V1.2$ current characteristics and the impact of the recording temperature, charge carrier, and current region where drug effects were quantified on pharmacology. Results show that $Ca_V1.2$ current exhibits prominent rundown following whole cell formation, was exquisitely temperature sensitive, and required a high degree of series resistance compensation to optimize voltage control. These characteristics meant that drug-independent changes in the current amplitude can be anticipated during long lasting pharmacology experiments, and laboratory-specific practices to deal with these changes can be sources of data variability. In addition, all experimental factors examined affected drug potency estimations, with drug-specific magnitude and direction of change. For $Ca_V1.2$ data intended to support risk prediction or clinical interpretation, normalizing laboratory-specific practices is essential to promote data reproducibility across laboratories—a pivotal step toward engendering confidence amongst regulators for applying these *in vitro* data in the decision-making process. To support data transparency, the original electrophysiology records, detailed cell culture procedure, and supplemental materials for the present study are available for download at: https://osf.io/g3msb/.

## Methods

### Cells

CHO cells stably transfected with $hCa_v1.2α$, $β_2$, and $α_2δ_1$ subunits (Charles River Laboratory; CT3004) were cultured at 5% $CO_2$ and 37°C, following passage in Ham's F12 media with L-glutamine nutrient mixture (Gibco #11765054) supplemented with 10% tetracycline-screened

fetal bovine serum (FBS) (Cytiva Hyclone SH30071.03T) and the following cell selection reagents: Blasticidin (0.01 mg/mL; Gibco #A1113903), Geneticin (G418; 0.25 mg/mL; Sigma G8168), Hygromycin (0.25 mg/mL; Sigma H0654), and Zeocin (0.40 mg/mL; Invitrogen #46–0509). Cells were seeded at low density and kept in culture for 4–7 days before seeding on glass coverslips for electrophysiology use. By the time cells were detached for seeding, they were fully confluent. Twenty-four to 48 hours prior to recording, cultures were washed with DPBS without Ca$^{2+}$ or Mg$^{2+}$ (Gibco #14190144), and then detached by applying Accutase (Sigma A6964) for 2 minutes. Cell suspensions of 30,000–40,000 cells/mL were added to 35 mm petri dishes containing 12 mm glass coverslips, in Ham's F12 media containing only 10% FBS. Cells were kept at 5% CO$_2$ and 37˚C until recording. For this cell line, the expressions of β$_2$ and α$_2$δ$_1$ subunits were constitutive, while that of the pore-forming α subunit required tetracycline induction. Three protocols were used for tetracycline induction to accommodate staff schedule. For the first protocol, cells were seeded late in the afternoon the day prior to recording. On the next day, 16–20 hours after seeding, 2.5 μg/mL tetracycline (Sigma T7660) was added to the petri dishes for 4 hours prior to recording. For the second protocol, cells seeded the day before recording were allowed to attach to glass coverslips for ~6 hours, and 0.5 μg/mL tetracycline was added for overnight induction (typically 16–20 hours prior to recording on the following day). For the third protocol, cells were seeded 4 days prior to recording. The day before recordings, cells were fully detached and seeded in media containing 1 μg/mL tetracycline. On the day of the recording, cells were detached again and seeded on glass coverslips. Regarding the first two protocols, after induction of the α subunit cells adopted a very flat morphology, rendering patching and maintaining long lasting recordings challenging. Cells generated using the third protocol were easier to patch due to the more rounded morphology. The use of different cell culture procedures did not impact pharmacology in this study. The amplitude of Ca$_V$1.2 current was dependent on both the amount of tetracycline used and the duration of induction.

## Electrophysiology

Voltage clamp recordings were made with Multiclamp 700B amplifier (Molecular Devices, CA) and digitized using a Digidata 1550B (Molecular Devices, CA) interface and the pClamp 10 software (Molecular Devices, CA). Glass coverslips with cells were placed in a recording chamber mounted on an inverted (Zeiss Axiovert 135TV or A1) or an upright microscope (Zeiss AxioExaminer D1), and the recording chamber was continuously perfused using a gravity-fed perfusion system, with an external solution flowing at a rate of 1.5–3 mL/min. Temperature of the recording solution was elevated using a dual channel temperature controller. Two controller models were used: 1) TC2BIP from Cell MicroControls, which elevated the solution temperature with an inline solution heater, and maintained the bath temperature by heating the ITO-coated glass coverslip which formed the bottom of the recording chamber; and 2) TC-344C from Warner Instruments, which elevated the solution temperature with an inline solution heater, and maintained the bath temperature by heating up the anodized aluminum platform (PH-1) that supported the edge of the plastic recording chamber. Two differences were noted regarding these two controller models: 1) at near 37˚C, ~2˚C difference between the inflow and center and the recording chamber was observed for TC-344C; and 2) TC2BIP provided more stable temperature control near 37ºC than TC-344C, even when the flow rate changed. Bath temperature in the recording chamber was recorded using a thermistor placed in the bath throughout the experiment. Whole-cell current was recorded at near physiological temperature (PT; 37 ± 2˚C) or room temperature (RT; 23 ± 1˚C). While most of the recordings in this study aimed at 37˚C, the term "near PT" was used to acknowledge the few degrees of temperature fluctuations that occurred during the experiments.

The internal solution contained (in mM): 120 aspartic acid, 120 CsOH, 10 CsCl, 10 EGTA, 5 MgATP, 0.4 TrisGTP, 10 HEPES; pH adjusted to 7.4 with 5M CsOH; ~290 mOsM. When Ca$^{2+}$ was used as the charge carrier, the external solution contained (in mM): 137 NaCl, 4 KCl, 1.8 CaCl$_2$, 1 MgCl$_2$, 10 HEPES, 10 glucose; pH adjusted to 7.4 with 5M NaOH; ~290 mOsM. When Ba$^{2+}$ was used as the charge carrier, the external solution contained (in mM): 137 NaCl, 4 KCl, 4 BaCl$_2$, 1 MgCl$_2$, 10 HEPES, 10 glucose; pH adjusted to 7.4 with 5M NaOH; ~290 mOsM. Recording electrodes were made by pulling borosilicate glass pipettes (BF150-86-10; Sutter Instrument, CA) with a P97 micropipette puller (Sutter Instruments, CA), and had tip resistances in the range of 1.5–2.5 MΩ when filled with the internal solution. The voltage command values were corrected for the 17 mV liquid junction potential (LJP) that resulted from using the above internal solution and Ca$^{2+}$-containing external solution at 37˚C, estimated using the PClamp 10 software. Given that voltage sensed by the membrane equals to voltage at the pipette minus the LJP (or $V_m = V_{pipette} − V_{LJP}$), to hold the cell at -80 mV, the input voltage was set at -63 mV. The 17 mV LJP correction was also applied to RT recordings using Ca$^{2+}$ as the charge carrier (LJP at 23˚C was estimated to be 16 mV), and to near PT recordings using Ba$^{2+}$ as the charge carrier (LJP at 37˚C was estimated to be 17 mV). The voltage waveform used was as follows: from a holding potential of -80 mV, the cell was hyperpolarized to -90 mV for 100 ms, repolarized to -80 mV for 100 ms, depolarized to 0 mV for 40 ms, further depolarized to +30 mV for 200 ms, and finally ramped down to -80 mV in 100 ms (-1.1 V/s). This waveform, modified from that used by Li et al., 2018 [6], was chosen as it evoked an inward current peak at the 0 mV step (where Li et al., 2018 characterized drug effects at) and another one at the repolarizing ramp that reflects the channel state of what Crumb et al., 2016 studied [5]. The use of this voltage protocol therefore permitted a direct comparison of drug effects at the step and the ramp current that were separated in time and associated with different membrane voltages/channel states. The voltage waveform was delivered at 5 s intervals or 0.2 Hz. Signals were filtered at 3 or 10 kHz and sampled at 10 kHz. Whole cell capacitance was neutralized. Series resistance (R$_s$) was measured approximately 2 minutes following whole cell formation, after signs of membrane resealing were no longer evident, using the membrane test function of the pClamp 10 software. R$_s$ was electronically compensated at 80%. The Multi-Clamp 700B R$_s$ compensation bandwidth control replaces the "lag" control on earlier Axon amplifier series: Bandwidth = 1 / (2 $^*$ π $^*$ Lag). This study used the default R$_s$ correction bandwidth of 1.02 kHz, which is equivalent to a lag value of 156 μs. For the pharmacology dataset in this manuscript, R$_s$ was 4.5 ± 0.1 MΩ (± SEM; n = 295; RT and near PT data combined); whole cell capacitance was 35.2 ± 0.9 pF (n = 294; value from one cell was not captured).

Ca$_V$1.2 current showed pronounced rundown in whole cell configuration (see "Results and discussion"). To study drug effects, after Ca$_V$1.2 current reached a quasi-steady state level in the control solution, drug solution was perfused as the recording continued. Depending on the cell quality and stability achieved in the control solution, 1 to 2 drug concentrations were tested per cell. Verapamil at 100 μM was used as a full blocker and was applied at the end of the recordings whenever possible.

## Drugs

Naloxone hydrochloride (0599), tolterodine L-tartrate (3761), and diltiazem hydrochloride (0685) were purchased from Tocris Bioscience. Buprenorphine hydrochloride (B9275, USDEA C-III), (±)-methadone hydrochloride (M0267, USDEA C-II), naltrexone hydrochloride (N3136), (±)-verapamil hydrochloride (V4629), and DMSO (D8418) were purchased from Sigma-Aldrich. Norbuprenorphine hydrochloride (USDEA C-II) was purchased from Noramco. Stock solutions of naloxone, naltrexone, and verapamil were dissolved in milliQ

water. Stock solutions of methadone, buprenorphine, norbuprenorphine, diltiazem, and tol-terodine were dissolved in DMSO. When DMSO was used as a solvent, the % of DMSO exposed to cells was $\leq$0.3%. Aliquoted stock solutions were stored at -20˚C until the day of experiments and were diluted to specific test concentrations in the external solution.

## Data analysis and reporting

Data analysis and curve fitting were done in Clampfit 10.6 (Molecular Devices, CA) and Igor Pro 8.0 (WaveMetrics). Two offline methods were used to isolate Ca$_V$1.2 current from the total inward current. The first method was the passive current (I$_{passive}$) subtraction. Ohm's law was used to calculate the resting input resistance (R$_{input}$) for each current trace:

$$R_{input} = (V_{-90\ mV} - V_{-80\ mV})/(I_{-90\ mV} - I_{-80\ mV})$$

Here I$_{-80\ mV}$ refers to the current measured at the holding potential of -80 mV (V$_{-80\ mV}$), and I$_{-90\ mV}$ the current measured during hyperpolarizing step to -90 mV (V$_{-90\ mV}$). I$_{passive}$ was defined to exhibit linear current-voltage (I-V) relationship. Therefore, assuming that R$_{input}$ was constant across all voltages, I$_{passive}$ was calculated using the following equation:

$$I_{passive} = I_{-80\ mV} + (V - V_{-80\ mV})/R_{input}$$

Here V refers to any voltage within the "step-step-ramp" protocol. Using the custom macros written for Igor Pro, I$_{passive}$ was calculated for each current trace and then subtracted from that trace to yield Ca$_V$1.2 current.

The second current subtraction method was verapamil subtraction. At positive membrane potentials, a population of cells exhibited a non-linear outward current that was not removed by I$_{passive}$ subtraction. This outward current was most notable at the +30 mV step and the adjoining repolarizing ramp section. For these cells, the residual current trace in the presence of 100 μM verapamil was subtracted from all current traces to isolate Ca$_V$1.2 current. Verapamil subtraction was performed by averaging several traces recorded in verapamil that exhibited full inhibition of the ramp current, and then subtracting this averaged trace from all recorded current traces. For the methadone experiments, some cells did not receive 100 μM verapamil. Nonetheless, complete elimination of the ramp current was achieved by 100 and 300 μM methadone. In these cases, several traces following full ramp current inhibition by methadone were averaged and then subtracted from all recorded traces to isolate Ca$_V$1.2 current. S1 Fig illustrates these two subtraction methods and the phenotype of cells to which each was applied. Of note, even when the ramp current was completely eliminated by verapamil or methadone, a sizable portion of the 0 mV step current remained (see "Results and discussions"). Therefore, the step current was always quantified from I$_{passive}$-subtracted traces. The ramp current was quantified using I$_{passive}$-subtracted traces when there was no overt outward current at the positive membrane potentials or using verapamil-subtracted (or methadone-subtracted) traces when there was overt outward current, and the recording showed no time-dependent changes in the passive membrane properties, inferred from stability of R$_{input}$ and I$_{-80\ mV}$.

From the I$_{passive}$- or verapamil-subtracted current traces, the step and the ramp current were measured as the most negative inward current at the 0 mV step and the entire repolarizing ramp, respectively. Fractional inhibition by the tested drug for each cell was calculated with the following equation:

$$\text{Fractional inhibition} = 1 - \left(I_{drug}/I_{control}\right)$$

Here $I_{drug}$ is the averaged current amplitude from the last 10 traces recorded in the drug concentration, and $I_{control}$ is the averaged current amplitude from the last 10 traces recorded in the control solution. Fractional inhibition values for individual cells were plotted against concentrations tested to yield concentration-inhibition graphs, and individual data points were fit with the Hill equation to estimate drug potency:

$$\text{Fractional inhibition} = 1/(1 + (IC_{50}/[drug])^{n_H})$$

Here $IC_{50}$ is the concentration that inhibited 50% of the current, [drug] is the drug concentration, and $n_H$ is the Hill coefficient. The upper and lower 95% confidence interval (CI) bands were also plotted in the concentration-inhibition graphs to demonstrate uncertainty of the fit parameters. Except for the $IC_{50}$ and $n_H$ values, all data are presented as mean ± SEM.

$IC_{50}$s of buprenorphine, norbuprenorpine, methadone, naltrexone, and naloxone on the ramp current studied in external $Ca^{2+}$ and at 37ºC were published previously [18]. The $IC_{50}$ values there differed slightly from those presented in this manuscript because they were estimated by fitting the averaged fractional inhibition values (instead of individual cells' values) at different concentrations.

A t-test with equal variance was performed to compare the extent of $Ca^{2+}$ and $Ba^{2+}$ current rundown, using Prism version 8.4 (GraphPad, CA).

## Results and discussion

### Rundown of $Ca_V1.2$ current in external $Ca^{2+}$ or $Ba^{2+}$ at near PT

Fig 1A and 1B show $Ca_V1.2$ current recorded in external $Ca^{2+}$ and $Ba^{2+}$, respectively. The maximal peak current evoked by the 0 mV step is referred to as $I_{Ca-step}$ or $I_{Ba-step}$ depending on the charge carrier used; the maximal ramp current evoked by the repolarizing ramp, $I_{Ca-ramp}$ or $I_{Ba-ramp}$. Fig 1C–1F show the time course plots of normalized current amplitudes recorded in the control solution and following verapamil (100 μM) application. After whole cell formation, $Ca_V1.2$ current showed prominent rundown regardless of which charge carrier was used, and in most cells eventually reached a quasi-steady state level. Current rundown was seen in every cell and was characterized by a fast phase followed by a slower phase. The normalized $I_{Ca-step}$ at the 150th trace was 0.42 ± 0.04 relative to the 1st trace (Fig 1C); the normalized $I_{Ca-ramp}$, 0.44 ± 0.06 (Fig 1D). The normalized $I_{Ba-step}$ was 0.34 ± 0.04 (Fig 1E); the normalized $I_{Ba-ramp}$, 0.32 ± 0.04 (Fig 1F). To assess whether the extent of rundown was different between the $Ca^{2+}$ and the $Ba^{2+}$ current, a basic statistical analysis of the differences in the amplitude of the 150th trace relative to the first trace was performed. A t-test with equal variance revealed no significant difference between $I_{Ca-step}$ and $I_{Ba-step}$ and between $I_{Ca-ramp}$ and $I_{Ba-ramp}$ ($p > 0.05$). Fig 1G and 1H show the time course of the ratios of ramp-to-step current for traces acquired in the control solution. For $Ca^{2+}$ current, the averaged ratio was 0.33 ± 0.03; for $Ba^{2+}$ current, 0.60 ± 0.05. The larger ratio obtained in $Ba^{2+}$ is consistent with the removal of $Ca^{2+}$-dependent inactivation [19, 20]. These ratios remained constant throughout the control recording despite current rundown, suggesting that rundown reflects a progressive loss in the available channels. Rundown assessed using the present voltage protocol at 0.2 Hz was not attributed to intracellular $Ca^{2+}$ accumulation, since it occurred to a similar degree in external $Ba^{2+}$.

Application of verapamil nearly eliminated the ramp current (Fig 1D and 1F) but not the step current (Fig 1C and 1E). Relative to the 150th trace recorded in the control solution, the residual $I_{Ca-step}$ in verapamil was 20 ± 2%, while that of $I_{Ca-ramp}$ was 4 ± 1%. Likewise, the residual $I_{Ba-step}$ in verapamil was 32 ± 4%, while that of $I_{Ba-ramp}$ was 2 ± 0%. Therefore, amplitude of the step current was always quantified using $I_{passive}$-subtracted traces, not verapamil-subtracted traces (see "Methods").

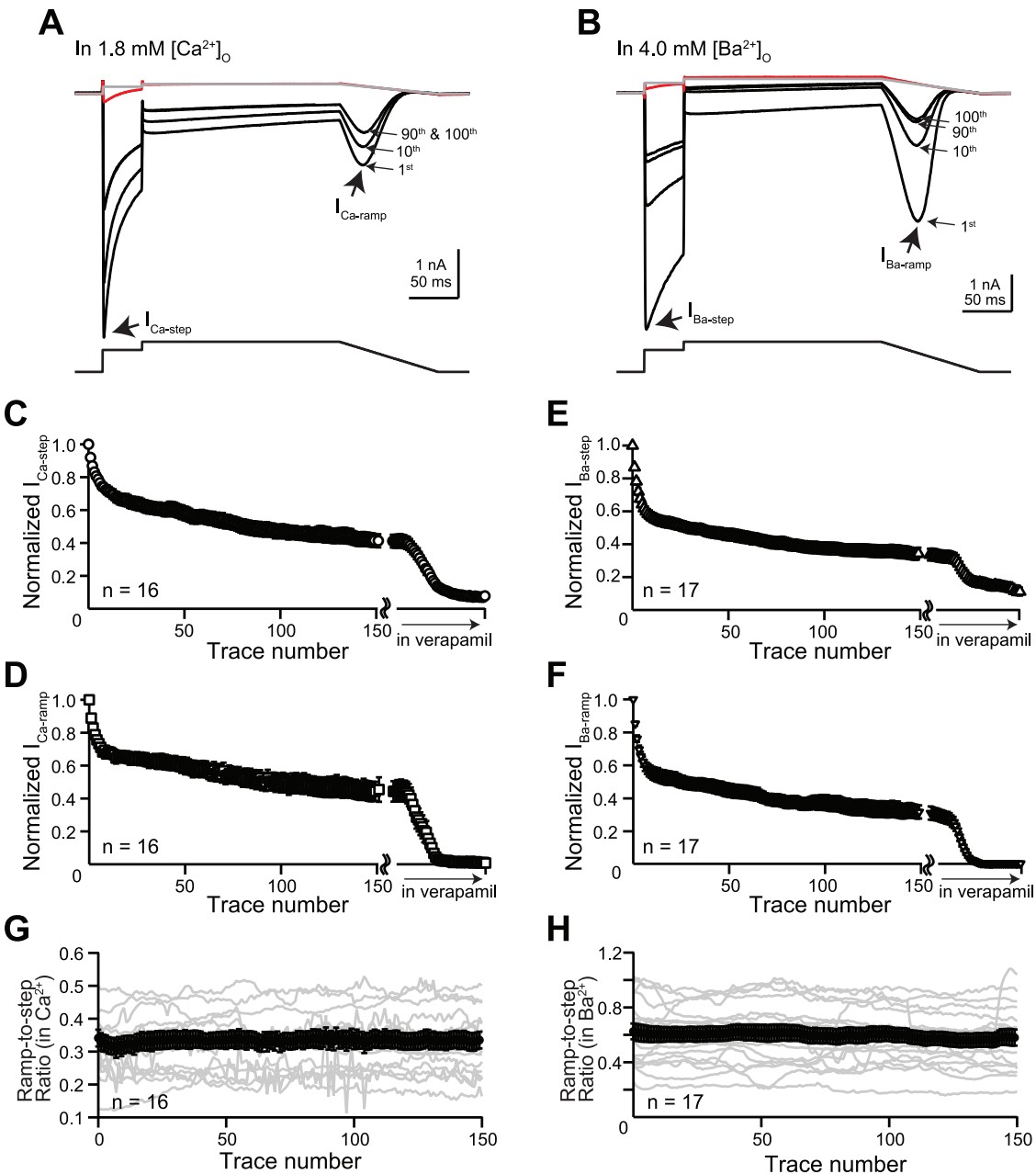

**Fig 1. Rundown of Ca_V1.2 channel activity at near PT. A, B**. *Top*, representative current traces from 2 cells recorded in either external $Ca^{2+}$ **(A)** or $Ba^{2+}$ **(B)**. Cell ID for **(A)** was 19327000; for **(B)**, 19404005. Black traces reflect recordings obtained in the control solution, and the $1^{st}$, $10^{th}$, $90^{th}$, and $100^{th}$ traces recorded after ~2 minutes of whole cell dialysis are shown. Red traces reflect recordings obtained following application of 100 μM verapamil, after steady state inhibition was achieved. $I_{passive}$ traces, shown in gray, were calculated using $R_{input}$ derived from the verapamil traces. *Bottom*, the voltage protocol used. **C, D)** Summary time course plots of normalized $I_{Ca-step}$ **(C)** and $I_{Ca-ramp}$ **(D)** in the control solution and following verapamil application (n = 16). Data points are shown as mean ± sem. Verapamil application did not start at the same time for every cell. Therefore, the X-axes show a break after trace 150 to synchronize the data points obtained following verapamil application. **E, F)** Summary time course plots of normalized $I_{Ba-step}$ **(E)** and $I_{Ba-ramp}$ **(F)** in the control solution and following verapamil perfusion (n = 17). **G, H)** Ratio of $I_{Ca-ramp}$-to-$I_{Ca-step}$ **(G)** or $I_{Ba-ramp}$-to-$I_{Ba-step}$ **(H)** for traces acquired in the control solution. Data for individual cells are shown as light gray lines. Mean ± sem are shown as black symbols plus error bars.

Rundown of Ca$_V$1.2 current in whole cell configuration is a widely observed phenomenon. Since rundown leads to overestimation of fractional inhibition, laboratory-specific tolerance regarding the rate of rundown and practices to correct rundown can introduce variable degrees of imprecision into drug potency estimation. The diverse practices are exemplified by the two publications that motivated the present study. Crumb et al., 2016 did not correct for rundown, stated that cells with >20% rundown were discarded, but did not define how many current traces the 20% calculation was based [5]. Li et al., 2018 used perforated population recording that presumably reduced rundown, yet still corrected for rundown in the calculation of drug inhibition [6]. Even if the rates of Ca$_V$1.2 current rundown were similar between the two publications, these different practices alone would lead to different drug potency estimations. Experience from this laboratory indicates that the rate of rundown, and whether current can reach a quasi-steady state in the control solution for pharmacology experiments are functions of the cell line used and cell culture conditions, and therefore unlikely to be the same across studies. Using the same Ca$^{2+}$-external solution, internal solution, and voltage protocol, two additional Ca$_V$1.2 cell lines that expressed the same channel subunits were tested, and both cell lines showed near complete loss of Ca$_V$1.2 current at near PT with time, without reaching a quasi-steady state level (S2 Fig). Can Ca$_V$1.2 current rundown under whole cell mode be prevented? Previous studies suggest that rundown of cardiac L-type Ca$^{2+}$ channel activity reflects channel dephosphorylation. This conclusion is based on evidence from native myocytes that manipulating protein kinase A (PKA)-mediated phosphorylation [21], protein phosphatase activity [21], and increasing the intracellular level of ATP or cyclic AMP—molecules that enhance PKA-mediated phosphorylation [22] all led to expected changes in Ca$^{2+}$ channel activity. In the present study, inclusion of 5 mM MgATP and 0.4 mM TrisGTP in the internal solution did not prevent current rundown in the three cell lines tested. Therefore, rundown in whole cell configuration could not be prevented by simply supplying ATP.

## Temperature sensitivity of Ca$_V$1.2 current

During near PT recordings, the step current sometimes showed amplitude fluctuations that occurred without changes in the passive membrane properties (i.e., R$_{input}$ or I$_{-80 \text{ mV}}$; S3 Fig). As these amplitude fluctuations were not observed with RT recordings, one hypothesis is that they are associated with temperature fluctuations during the near PT recordings. Therefore, temperature sensitivity of the Ca$_V$1.2 current was examined. These experiments were conducted using setups with the TC-344C controller, as amplitude fluctuations were more common when recorded using these setups, and the thermistor measuring the bath temperature was positioned as close to the recorded cell as possible. After the current reached a quasi-steady state level at near PT, temperature control of the aluminum platform was turned off to allow graded bath temperature drop at the recording chamber, and then back on to elevate the bath temperature. Fig 2A shows the time course plots of I$_{Ca-step}$ and I$_{Ca-ramp}$ from a representative cell. The boxed regions are expanded and shown in Fig 2B. I$_{Ca-step}$ decreased and increased as the bath temperature lowered and elevated, respectively (*top*; note that Ca$^{2+}$ current amplitudes are expressed as negative values), while I$_{Ca-ramp}$ appeared to show the opposite pattern (*bottom*). No change in R$_{input}$ or I$_{-80 \text{ mV}}$ was observed (Fig 2C). The left panel of Fig 2D shows the plot of I$_{Ca-step}$ vs. temperature for this cell, demonstrating a strong linear relationship (r = -0.94). In contrast, no relationship was observed between I$_{Ca-ramp}$ and temperature (r = 0.38; Fig 2D, *right*). Temperature sensitivity of the Ca$^{2+}$ current was confirmed in 5 more cells (S4A–S4E Fig). In all cells recorded, I$_{Ca-step}$ showed strong linear relationship with temperature (r = -0.77 to -0.94). The I$_{Ca-ramp}$-temperature relationship was inconsistent: 4 cells showed no relationship (Fig 2D, *right*; S4A, S4B and S4D Fig., *right*), and 2 showed linear relationship

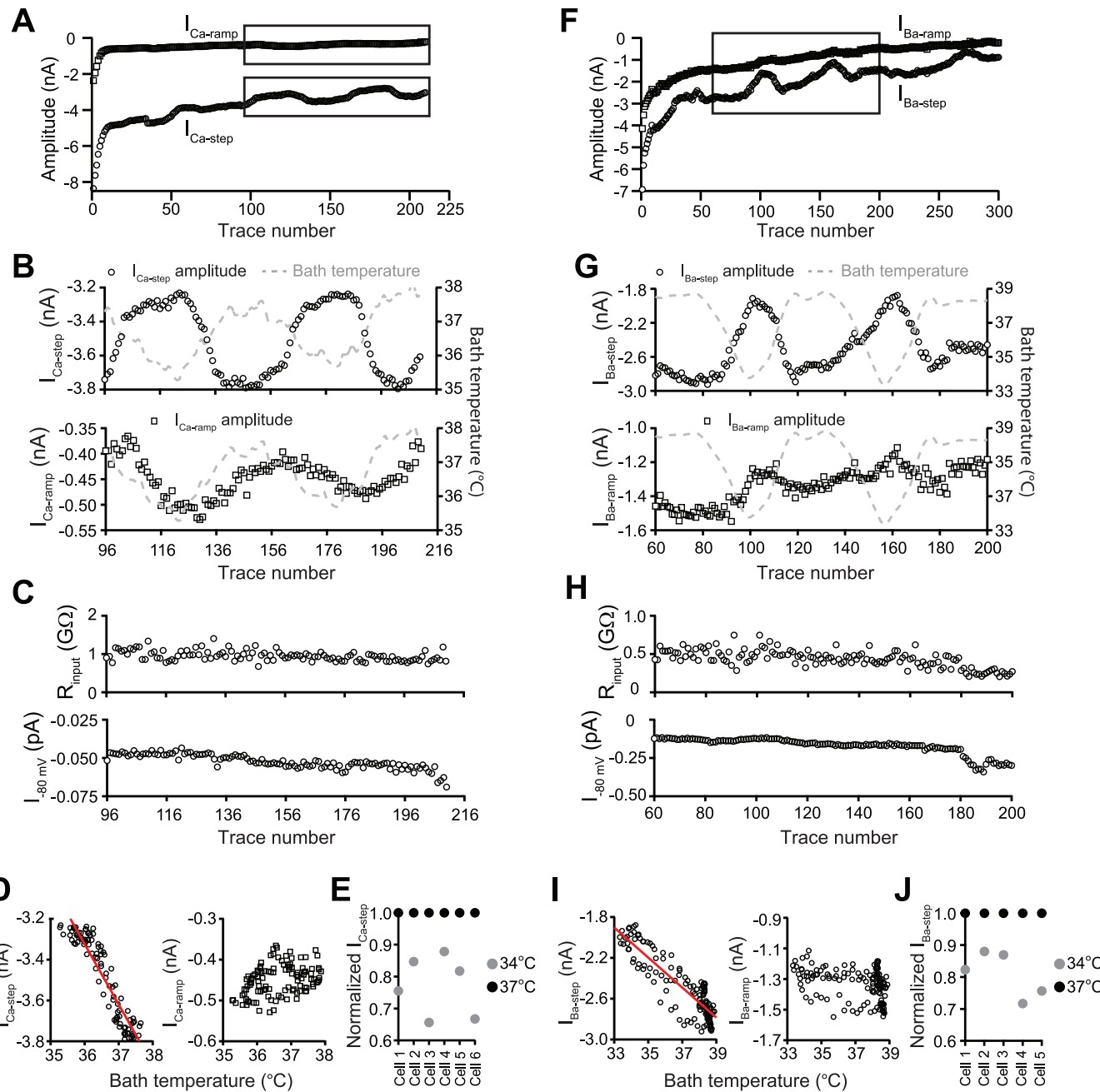

**Fig 2. Effects of recording temperature on Ca$_V$1.2 channel activity.** Cell ID for **(A)** through **(D)** was 19318007. Recordings were obtained in external Ca$^{2+}$.
**A)** Time course plots of I$_{Ca-step}$ (open circle) and I$_{Ca-ramp}$ (open square) for the entire experiment. The boxed regions were expanded and shown in **(B)**. **B)** *Top*, time course plots of I$_{Ca-step}$ from traces 96 to 216 and bath temperature (gray dotted line). Current amplitudes were corrected for rundown. This could be done since rundown correction was performed on the traces used for the fit. To estimate rundown, data points in **(B)** were fit with a linear function to yield a slope of 0.00495 nA/trace. This amount of current loss was then added back to the original I$_{Ca-step}$ amplitudes. *Bottom*, time course plots of I$_{Ca-ramp}$ and bath temperature. Current amplitudes were also corrected for run rundown (slope = 0.00146 nA/trace). **C)** R$_{input}$ (*top*) and I$_{-80\ mV}$ (*bottom*). **D)** *Left*, rundown-corrected I$_{Ca-step}$ vs. bath temperature. These data points were fit with a linear function, yielding a slope of -0.295 nA/˚C and a Y-intercept of 7.299 nA (r = -0.94). *Right*, rundown-corrected I$_{Ca-ramp}$ vs. bath temperature (r = 0.38). **E)** Calculated and normalized I$_{Ca-step}$ at 37˚C (black circle) and 34˚C (gray circle). Cell ID for **(F)** to **(J)** was 07_07_0008. Recordings were obtained in external Ba$^{2+}$. **F)** Time course plots of I$_{Ba-step}$ (open circle) and I$_{Ba-ramp}$ (open square) for the entire experiment. The boxed region was expanded and shown in **(G)**. **G)** Time course plots of I$_{Ba-step}$ (*top*) and I$_{Ba-ramp}$ (*bottom*) from traces 60 to 200 and bath temperature (gray dotted line). Both I$_{Ba-step}$ and I$_{Ba-ramp}$ were corrected for rundown, using 0.00741 nA/trace and 0.00538 nA/trace, respectively. **H)** R$_{input}$ (*top*) and I$_{-80\ mV}$ (*bottom*). **I)** Rundown-corrected I$_{Ba-step}$ (*left*) or I$_{Ba-ramp}$ (*right*) vs. bath temperature. Fitting I$_{Ba-step}$ vs. temperature plot with a linear function yielded a slope of -0.147 nA/˚C and a Y-intercept of 2.947 nA (r = -0.88). Rundown-corrected I$_{Ba-ramp}$ vs. temperature plot (r = -0.36). **J)** Calculated and normalized I$_{Ba-step}$ at 37˚C (black circle) and 34˚C (gray circle).

(r = -0.73 to -0.88). Fig 2E shows the calculated and normalized $I_{Ca-step}$ at 37˚C and 34˚C. Depending on the cell, 3˚C of temperature drop reduced $I_{Ca-step}$ amplitude by 12% to 34%. This magnitude of temperature fluctuation was routinely observed during near PT experiments for setups with TC-344 controller.

Temperature sensitivity of the $Ba^{2+}$ current was examined as well. Fig 2F shows the time course plots of $I_{Ba-step}$ and $I_{Ba-ramp}$ from a representative cell, and the boxed region was expanded and shown in Fig 2G. $I_{Ba-step}$ changed in the same direction as the bath temperature (*top*), and $I_{Ba-ramp}$ seemed to not respond to temperature changes (*bottom*). $R_{input}$ and $I_{-80 mV}$ remained stable during temperature manipulations (Fig 2H). The left panel of Fig 2I shows the plot of $I_{Ba-step}$ vs. temperature, demonstrating a clear linear relationship (r = -0.88). The plot of $I_{Ba-ramp}$ vs. temperature, on the other hand, did not reveal any relationship (r = -0.36; Fig 2I, *right*). Temperature-dependent effect on the $Ba^{2+}$ was confirmed in 4 more cells (S4F–S4I Fig). In all cells recorded, $I_{Ba-step}$ showed strong linear relationship with temperature (r = -0.87 to -0.93). The data for $I_{Ba-ramp}$ and temperature were inconsistent: 3 cell showed no-to-weak relationship (Fig 2I, *right*; S4F and S4G Fig, *right*), and 2 showed strong relationship (r = -0.80 and -0.88). Fig 2J shows the calculated and normalized $I_{Ba-step}$ at 37˚C and 34˚C. Depending on the cell, 3˚C temperature drop reduced $I_{Ba-step}$ by 18% to 28%.

The step current in the present study is very temperature-sensitive regardless of which charge carrier was used. These results are consistent with prior studies conducted on cloned Ca<sub>V</sub>1.2 channels comprised of $\alpha_{1c}$, $\alpha_2/\delta_a$, and $\beta_{1b}$ (or $\beta_{2c}$) expressed in xenopus oocytes [23] and L-type $Ca^{2+}$ channels in native ventricular myocytes [24, 25] that also showed high temperature sensitivity. Increasing PKA-mediated phosphorylation reduced temperature sensitivity of some $Ca^{2+}$ channel gating parameters, suggesting that the high temperature sensitivity may be due to channels in the unphosphorylated state [25]. Temperature sensitivity of the step current adds an additional challenge to conducting pharmacology experiments at near PT, since several degrees of temperature fluctuations can easily be produced by a slowing of the flow rate due to the presence of bubbles in the perfusion line and/or reduced fluid level in the reservoirs of the gravity-fed perfusion system. Temperature sensitivity of the Ca<sub>V</sub>1.2 current is thus a source of variability for pharmacology data generated within and across laboratories.

## Consequence of $R_s$ compensation on Ca<sub>V</sub>1.2 current at near PT

The cells used to generate Fig 1 were used to estimate the activation kinetics and amplitude of Ca<sub>V</sub>1.2 current at near PT. $I_{Ca-step}$ reached peak 1.84 ± 0.17 ms following the voltage jump to 0 mV (range: 1.34 to 3.39 ms; n = 12, using cells with clear separation between capacitive transient and the step current), and was -2210.1 ± 178.3 pA in amplitude (range: -1376.7 to -3362.9 pA, based the average of 91st to 100th traces recorded in the control solution). $I_{Ca-ramp}$ reached peak at 0.6 ± 0.6 mV during the repolarizing ramp (range: -4.0 to 3.6 mV; n = 16), and was -814.0 ± 108.4 pA in amplitude (range: -320.2 to -1546.3 pA). The fast kinetics and relatively large amplitudes of the step current prompted for a set of experiments to assess the impact of $R_s$ compensation in these recordings. During these experiments, $R_s$ was measured and compensated at 80% before the start of recording. After Ca<sub>V</sub>1.2 current reached a quasi-steady state level in the control solution, $R_s$ compensation was turned "off" and "on" as the recording continued. Fig 3A and 3B show representative current traces and time course plots of $I_{Ca-step}$ and $I_{Ca-ramp}$ from one cell obtained with and without $R_s$ compensation. The total initial $R_s$ of this cell was 5.6 MΩ. With $R_s$ compensation, $I_{Ca-step}$ was ~600 pA larger than without $R_s$ compensation (Fig 3B, *top*). In contrast, no difference was observed in $I_{Ca-ramp}$ amplitude (*bottom*) or the ramp voltage at which $I_{Ca-ramp}$ reached peak (Fig 3C, *top*). $R_{input}$ and $I_{-80 mV}$ remained stable (Fig 3C, *middle and bottom*). The impact of $R_s$ compensation was confirmed in 5

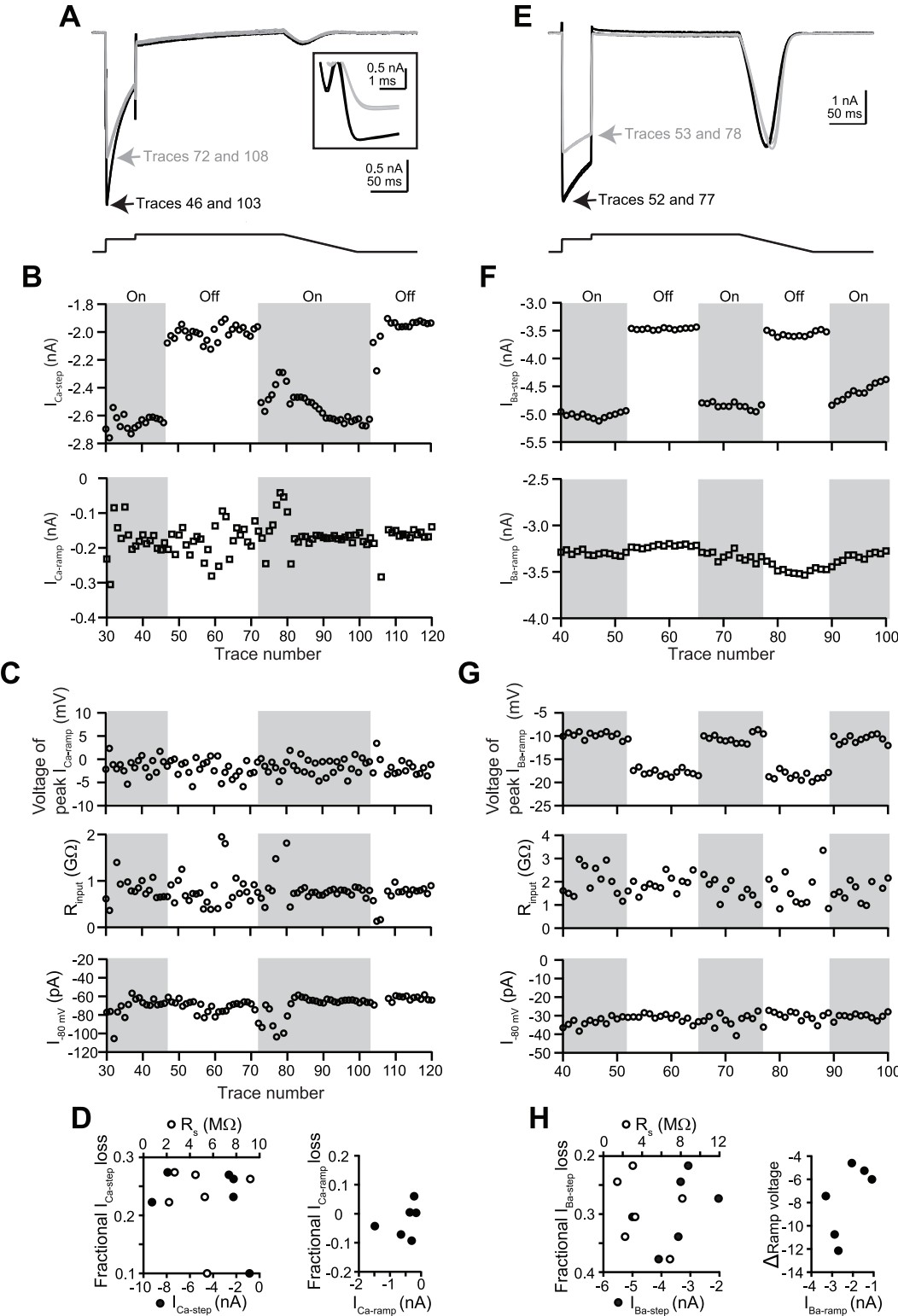

**Fig 3. Effects of R$_s$ compensation on Ca$^{2+}$ and Ba$^{2+}$ currents.** Data shown in panels (**A**) through (**D**) were obtained in external Ca$^{2+}$. Cell ID was 21318008 for panels (**A**) through (**D**). **A)** *Top*, representative Ca$^{2+}$ current traces obtained from a cell, with (black traces) and without R$_s$ compensation (gray traces). *Bottom*, the voltage protocol used. **B)** Time course plots of I$_{Ca-step}$ (*top*) and I$_{Ca-ramp}$ (*bottom*), focusing on traces 30 to 120 for which R$_s$ compensation was turned on and off. No rundown correction was made for these plots. **C)** Time course plots of ramp voltage at which I$_{Ca-ramp}$ reached peak

amplitude (*top*), $R_{input}$ (*middle*), and $I_{-80\ mV}$ (*bottom*) for traces 30 to 120. **D**) *Left*, fractional $I_{Ca-step}$ loss due to not compensating for $R_s$, calculated as the ratio of $I_{Ca-step}$ without $R_s$ compensation vs. $I_{Ca-step}$ with 80% $R_s$ compensation, and plotted against $I_{Ca-step}$ obtained with $R_s$ compensation (lower x-axis) or the amount of $R_s$ compensated (upper x-axis). *Right*, fractional $I_{Ca-ramp}$ loss due to not compensating for $R_s$, plotted against $I_{Ca-ramp}$ with $R_s$ compensation. Data shown in panels (**E**) through (**H**) were obtained in external $Ba^{2+}$. Cell ID was 2021_06_25_0004 for panels (**E**) through (**I**). **E**) Representative $Ba^{2+}$ current traces obtained from a cell, with (black traces) and without $R_s$ compensation (gray traces). **F**) Time course plots of $I_{Ba-step}$ (*top*) and $I_{Ba-ramp}$ (*bottom*), focusing on traces 40 to 100 during which $R_s$ compensation was turned on and off. No rundown correction was made for these plots. **G**) Time course plots of ramp voltage (*top*), $R_{input}$ (*middle*), and $I_{-80\ mV}$ (*bottom*) for traces 40 to 100. **H**) *Left*, fractional $I_{Ba-step}$ loss due to not compensating for $R_s$, plotted against $I_{Ba-step}$ obtained with $R_s$ compensation (lower x-axis) or the amount of $R_s$ compensated (upper x-axis). *Right*, delta (Δ) ramp voltage shift for $I_{Ba-ramp}$ due to not compensating for $R_s$, plotted against $I_{Ba-ramp}$ with $R_s$ compensation.

additional cells. The left panel of Fig 3D shows the fractional $I_{Ca-step}$ loss without $R_s$ compensation for all 6 cells, plotted against either $I_{Ca-step}$ with $R_s$ compensation (solid symbols) or the amount of $R_s$ that was compensated (or 80% total $R_s$; open symbols). No relationship was evident amongst these parameters, as expected since voltage loss through $R_s$ (hence fractional current loss) is a product of both the current amplitude and $R_s$. For these cells, voltage loss through $R_s$ at $I_{Ca-step}$ ranged from 4.7 to 21.3 mV, calculated by using Ohm's law. The right panel of Fig 3D shows the fractional $I_{Ca-ramp}$ loss due to not compensating for $R_s$, plotted as a function of $I_{Ca-ramp}$ with $R_s$ compensation. Collectively, no loss was observed (average: -0.02 ± 0.02).

The kinetics and amplitude of $Ba^{2+}$ current at near PT was also quantified. $I_{Ba-step}$ reached peak 2.91 ± 0.25 ms following the voltage jump to 0 mV (range: 1.64 to 4.23 ms; n = 11), and was -4147.6 ± 813.1 pA in amplitude (range: -1420.0 to -8395.0 pA). $I_{Ba-ramp}$ reached peak at -7.7 ± 0.7 mV (range: -2.5 to -10.5 mV; n = 11) and was -2077.7 ± 309.5 pA in amplitude (range: -555.2 to -3506.4 pA). Fig 3E showed representative $Ba^{2+}$ current traces obtained from a cell, with and without $R_s$ compensation. The total $R_s$ for this cell was 4.1 MΩ. Similar to $Ca^{2+}$ current recordings, $R_s$ compensation affected the amplitude of $I_{Ba-step}$ (by ~1500 pA; Fig 3F, *top*) but not the amplitude of $I_{Ba-ramp}$ (Fig 3F, *bottom*). However, the ramp voltage at which $I_{Ba-ramp}$ peaked was consistently shifted rightward when $R_s$ was not compensated, toward the more hyperpolarized potential (Fig 3G, *top*). No change in $R_{input}$ or $I_{-80\ mV}$ was observed (Fig 3G, *middle and bottom*). The impact of $R_s$ compensation on the $Ba^{2+}$ current was verified in 5 more cells. The left panel of Fig 3H shows the fractional $I_{Ba-step}$ loss due to not compensating for $R_s$, plotted against $I_{Ba-step}$ amplitude with $R_s$ compensation (solid symbols) or the amount of $R_s$ compensated (open symbols). No relationship was evident amongst these parameters. Voltage loss through $R_s$ at $I_{Ba-step}$ ranged from 4.8 to 28.2 mV. The right panel of Fig 3H summarizes the hyperpolarizing shift of the ramp voltage for $I_{Ba-ramp}$ without $R_s$ compensation. On average, the shift was -7.7 ± 1.3 mV (range: -4.6 to -12.1 mV) and was not accompanied by $I_{Ba-ramp}$ amplitude change (fractional $I_{Ba-ramp}$ loss: 0.03 ± 0.00). Using normalized I-V relation generated from 15 cells, the reversal potential of $Ba^{2+}$ current was estimated to be +35 mV (S5A Fig). The increase in $Ba^{2+}$ driving force due to the hyperpolarizing shift in the ramp voltage offers an explanation as to why $I_{Ba-ramp}$ amplitude was unaltered by not compensating for $R_s$, since this could compensate for the fewer number of $Ca_V1.2$ channels that would be open due to not clamping the membrane potential at the expected levels.

These results demonstrate the impact of $R_s$ compensation when recording fast and large $Ca^{2+}$ and $Ba^{2+}$ currents at near PT. While $R_s$ compensation is a good practice for voltage clamp experiments, can $R_s$ affect pharmacology results? A recent study that modeled drug block of $Na^+$ channels indicates so, with the degree of rightward shift of the concentration-inhibition graphs dependent on the magnitudes of $R_s$ and $Na^+$ current [26]. $R_s$ compensation is a practice that also differed between Crumb et al., 2016 (compensated) [5] and Li et al., 2018 (did not

compensate) [6] (Table 1). Therefore, the decision to apply $R_s$ compensation or not during voltage clamp experiments to characterize drug effects may also be a contributing factor inter-laboratory data variability.

## Effects of recording temperature on verapamil and methadone inhibition of $Ca_V1.2$ current

The preceding sections focused on the $Ca^{2+}$ and $Ba^{2+}$ current characteristics at near PT. The next two sections focus on the impact of specific experimental factors on $Ca_V1.2$ channel pharmacology. The effect of recording temperature on $Ca_V1.2$ channel block was examined for verapamil and methadone. Fig 4A and 4C show representative recordings of $Ca^{2+}$ current obtained in the control solution and following verapamil application at near PT and RT, respectively. Fig 4B and 4D show concentration-inhibition plots of verapamil for $I_{Ca-step}$ and $I_{Ca-ramp}$ at these temperatures. At near PT, the $IC_{50}$ of verapamil inhibition of $I_{Ca-step}$ was 0.9 μM. This is 2.5X the $IC_{50}$ of $I_{Ca-ramp}$, which was 0.4 μM. The more potent inhibition at $I_{Ca-ramp}$ suggests continued block development throughout the 0 and the +30 mV steps that led to fewer channels available to reactivate during the repolarizing ramp. Since Cav1.2 channels enter inactivation following activation by the 0 mV step, a more potent inhibition of the ramp current relative to the step current using the present voltage protocol suggests inactivated state block in addition to the open state block (inferred by inhibition of the step current). At RT, the $IC_{50}$s of verapamil for $I_{Ca-step}$ and $I_{Ca-ramp}$ were 1.5 and 1.6 μM, respectively. The equally potent inhibition of the step and the ramp current at RT suggests a preference for open channel block. A surprisingly finding was that the slope of the concentration-inhibition relationship was temperature-dependent. At near PT, the $n_H$ values for $I_{Ca-step}$ and $I_{Ca-ramp}$ were 0.4 and 0.5, respectively. At RT, these values were 1 (Fig 4B and 4D). Assuming a single binding site, a $n_H$ value of 1 as observed at RT indicates a tight coupling between verapamil binding and block of $Ca^{2+}$ permeation, consistent with a direct pore blocking mechanism. On the other hand, the $n_H$ values of 0.4 and 0.5 as observed at near PT suggests a weaker coupling between verapamil binding and effect at the channel pore. This may occur if verapamil inhibition at near PT involves allosteric mechanisms (i.e., drug binding does not directly block $Ca^{2+}$ permeation, but rather induces channel closure or locking in the inactivated state). Verapamil inhibition of the $Ba^{2+}$ current was also studied at near PT (Fig 4E and 4F). Relative to the drug effect on the $Ca^{2+}$ current at near PT, the difference between the $IC_{50}$s of the step and the ramp current was more pronounced in external $Ba^{2+}$. The $IC_{50}$ at $I_{Ba-step}$ was 1.5 μM, 5X the $IC_{50}$ of $I_{Ba-ramp}$ which was 0.3 μM. The faster block development at the 0 and +30 mV step in external $Ba^{2+}$ could be due to verapamil interactions with either open and/or inactivated channel state, since more channels remained open at these voltages due to removal of $Ca^{2+}$-dependent inactivation. The $n_H$ values were low also, 0.3 and 0.5, respectively, similar to the drug effect on the $Ca^{2+}$ current at near PT and inconsistent with a direct pore block mechanism.

The consequence of recording temperature was also assessed for methadone on the $Ca^{2+}$ current. Fig 4G through Fig 4J show representative cells and corresponding concentration-inhibition plots at near PT and RT, respectively. At near PT, methadone exhibited complex effects on $I_{Ca-step}$. Collectively, facilitation was seen at 30 and 60 μM, and inhibition was seen at higher concentrations. The facilitatory effect was characterized by an increase in the peak amplitude, accelerated current decay during the 0 mV step (Fig 4G), and was specific for $I_{Ca-step}$ (Fig 4H). The latter point is clearly illustrated in Fig 4G: as $I_{Ca-step}$ showed a slight increase in 30 μM methadone, $I_{Ca-ramp}$ was nearly completely inhibited (Fig 4G). Given the complex effects at $I_{Ca-step}$, methadone's $IC_{50}$ was estimated for $I_{Ca-ramp}$ only, which was 5.4 μM with $n_H$ of 1.4. At RT, only inhibitory effect was observed. The $IC_{50}$ for $I_{Ca-step}$ at RT was 10.7 μM with

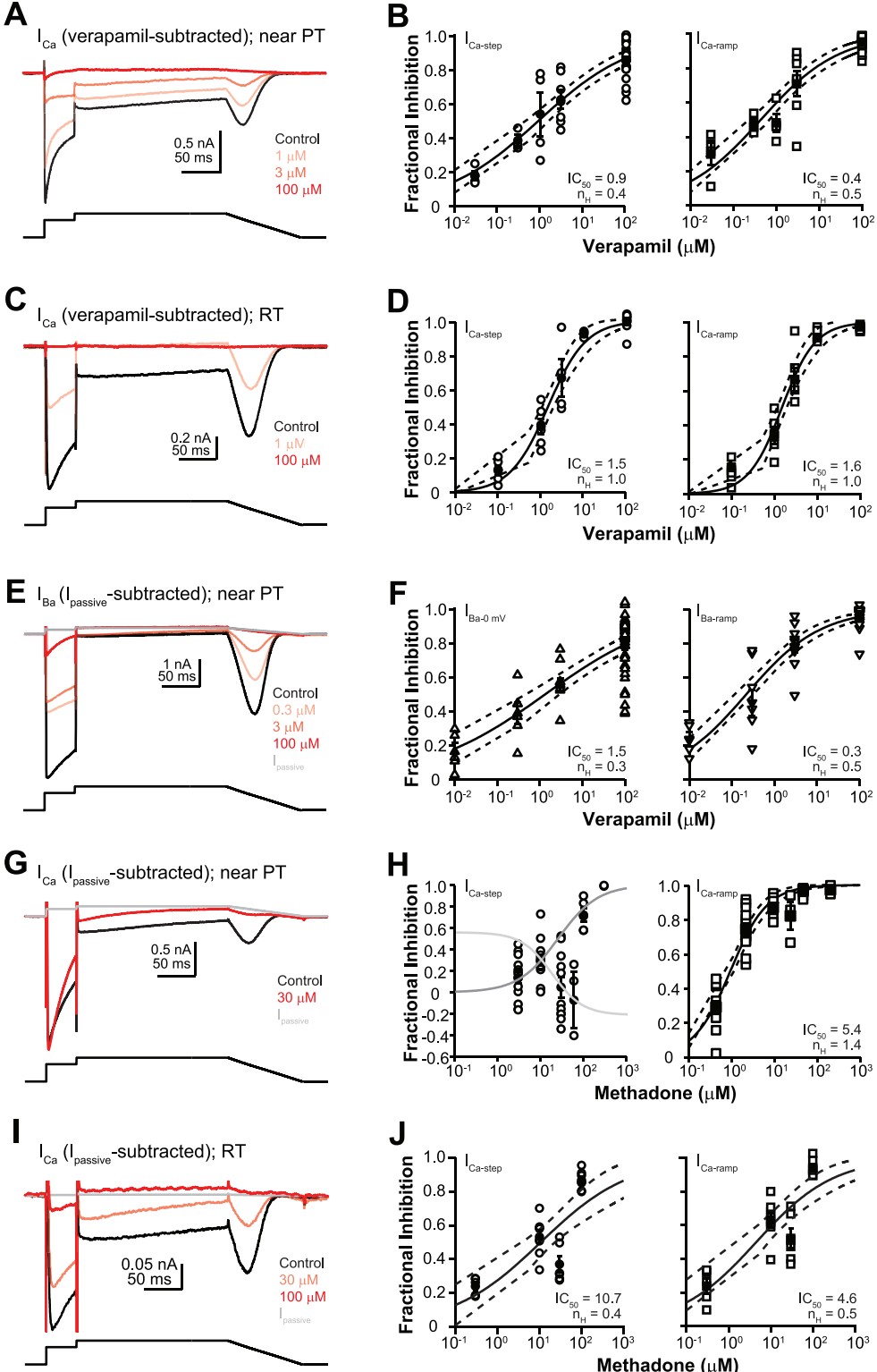

**Fig 4. Ca$_V$1.2 channel block by verapamil and methadone at near PT and RT. A)** Example current traces recorded from one cell (cell ID: 19820002) in external Ca$^{2+}$ at near PT. The illustrated traces were obtained in the control solution (black; trace 99), and following application of 1 μM (light red, trace 160), 3 μM (medium red, trace 209), and 100 μM verapamil (dark red; trace 250). This cell exhibited an outward current that was unmasked by 100 μM verapamil. Thus, Ca$_V$1.2 current was isolated using the verapamil-subtraction method. **B)** Concentration-inhibition

plots for $I_{Ca-step}$ (*left*) and $I_{Ca-ramp}$ (*right*) at near PT. Data points from individual cells were shown in open symbols ($I_{Ca-step}$, circle; $I_{Ca-ramp}$, square). Group averages (± sem) for different concentrations were shown in solid symbols plus error bars. The solid sigmoidal curve indicates the fit using the Hill equation; the dashed curves, upper and lower limit of the 95% CI of the fit. **C)** Example current traces recorded from one cell (cell ID: 19d31003) in external $Ca^{2+}$ at RT. The illustrated traces were obtained in the control solution (black; trace 54), and following application of 1 μM (light red, trace 110) then 100 μM verapamil (dark red; trace 155). This cell also exhibited an outward current that was unmasked by 100 μM verapamil. Thus, $Ca_V1.2$ current was isolated by using verapamil-subtraction method. **D)** Concentration-inhibition plots for $I_{Ca-step}$ (*left*) and $I_{Ca-ramp}$ (*right*) at RT. **E)** Example current traces recorded from one cell (cell ID: 19d04004) in external $Ba^{2+}$ at near PT. The illustrated traces were obtained in the control solution (black; trace 82), and following application of 0.3 μM (light red, trace 171), 3 μM (medium red, trace 228), and 100 μM verapamil (dark red; trace 286). $I_{passive}$ (gray) was calculated based on $R_{input}$ derived from trace 286. This cell had little to no outward current in the presence of verapamil, and $Ca_V1.2$ current was isolated using $I_{passive}$-subtraction method. **F)** Concentration-inhibition plots for $I_{Ba-step}$ (*left*) and $I_{Ba-ramp}$ (*right*) at RT. **G)** Example current traces recorded from one cell (cell ID: 19508007) in external $Ca^{2+}$ at near PT. The illustrated traces were obtained in the control solution (black; trace 115), and following application of 30 μM methadone (red, trace 196). $I_{passive}$ (gray) was calculated based on $R_{input}$ derived from trace 196. Note that in the presence of methadone, $I_{Ca-step}$ amplitude was slightly increased while its decay was accelerated. **H)** Concentration-inhibition plots for $I_{Ca-step}$ (*left*) and $I_{Ca-ramp}$ (*right*) at near PT. For visual presentation only, individual data points at 10, 30, and 60 μM reflecting methadone's facilitatory effect were fit with the Hill equation (light gray curve). To illustrate the inhibitory effect on $I_{Ca-step}$, individual data points excluding 30 and 60 μM methadone were fit with the Hill equation (dark gray curve). **I)** Example traces of current recorded from one cell (cell ID: NTCell_2019_08_29_0013) in external $Ca^{2+}$ at RT. The illustrated traces were obtained in the control solution (black; trace 100), and following application of 30 (medium red, trace 159) and 100 μM methadone (red, trace 250). $I_{passive}$ (gray) was calculated based on $R_{input}$ derived from trace 250. **J)** Concentration-inhibition plots of methadone's effects at RT.

$n_H$ of 0.4, and that for $I_{Ca-ramp}$ was 4.6 μM with $n_H$ of 0.5. These results suggest that methadone blocks $Ca_V1.2$ channels in both open and inactivated states at RT. This conclusion is consistent with a previous study that reported $IC_{50}$s of 26.6 μM for tonic or open channel block, and 7.7 μM for phasic or inactivated channel block, respectively, for methadone at ambient temperature [27]. In summary, methadone has dual effects at PT but not RT. Like verapamil, the $n_H$ value for methadone's inhibitory effect is temperature-dependent. Dual effects of drugs on L-type $Ca^{2+}$ channels in native myocytes that depend on the membrane potential have also been reported for dihydropyridine (+)-202-791 [28] and nitrendipine [29]. For nitrendipine, the concentration-facilitation plot clearly illustrated an inflection point, suggesting the existence of two binding sites with different affinities on the $Ca^{2+}$ channels [29]. Methadone used in the present study is a racemic mixture. Another study has reported that the R- and S-enantiomers of the cyclin-dependent kinase inhibitor roscovitine bind to different sites on $Ca_V1.2$ channels to affect activation and inactivation separately [30]. It is tempting to reconcile the present results by proposing distinct binding sites for methadone on $Ca_V1.2$ channels that are accessible at near PT but not RT. Follow-up studies are required to test this possibility.

A few drugs have been shown to exhibit temperature-dependent block on $Ca_V1.2$ channels in overexpression cells [20] or L-type $Ca^{2+}$ channels in native myocytes [21]. One study showed that nitrendipine and diltiazem inhibited $Ca^{2+}$ current mediated by $Ca_V1.2$, $\beta_1$, $\alpha_2/\delta$, and γ subunits more potently at RT than at 33°C [7]. Another study reported that increasing the recording temperature from 22°C to 37°C increased the block potencies of flavoxate and nifedipine on L-type $Ca^{2+}$ channels by 2.2X and 7X, respectively [8]. The direction and magnitude of potency shift due to temperature is thus drug-specific. Results of verapamil and methadone from the present study further extend those in the literature, demonstrating that recording temperature is an experimental factor that impacts $Ca_V1.2$ pharmacology.

## Comparisons of drug effects on the $Ca^{2+}$ and $Ba^{2+}$ currents at near PT

The effects of buprenorphine, norbuprenorphine, naloxone, and diltiazem were studied at near PT on the $Ca^{2+}$ and $Ba^{2+}$ currents; of naltrexone and tolterodine, on the $Ca^{2+}$ current

alone. Fig 5 shows the concentration-inhibition plots for these drugs. Fig 6 summarizes the IC$_{50}$s and n$_H$ values for all drugs studied. For near PT recordings, verapamil, buprenorphine, naloxone, diltiazem, and tolterodine inhibited I$_{Ca-ramp}$ more potently than I$_{Ca-step}$, suggesting that these drugs all have affinity for open and inactivated channels. Tolterodine showed the largest I$_{Ca-ramp}$ vs. I$_{Ca-step}$ IC$_{50}$ difference, by a factor of 8.7, suggesting a stronger preference for the inactivated state comparing with other drugs. For norbuprenorphine, there was no difference in the IC$_{50}$s for I$_{Ca-step}$ and I$_{Ca-ramp}$, suggesting a preference for open channel block when Ca$^{2+}$ was used as the charge carrier. Naltrexone was the only drug tested that showed higher IC$_{50}$ for I$_{Ca-ramp}$ than for I$_{Ca-step}$. This drug produced a dramatic concentration-dependent hyperpolarizing shift in the ramp voltage (Fig 7A), by -22 mV at 10 mM (Fig 7B), demonstrating an effect on voltage-dependence of channel gating. Using I-V generated from 14 cells, the reversal potential for Ca$^{2+}$ current under the current experimental condition was estimated to be +46 mV (S5B Fig). The lower fractional inhibition of I$_{Ca-ramp}$ relative to I$_{Ca-step}$ thus may not indicate drug unbinding during the +30 mV step. Instead, the increased driving force through channels that are available at more hyperpolarized membrane voltages in the presence of naltrexone may also be an explanation of the higher IC$_{50}$ for I$_{Ca-ramp}$ than for I$_{Ca-step}$. S6 and S7 Figs provide time course plots of individual cells tested with select drugs in Ca$^{2+}$ and Ba$^{2+}$, respectively.

These results demonstrate that even when drug effects were analyzed within the same cell using the same traces, different drug potencies could be obtained depending on which current region was analyzed. Similar analyses have been performed for (-)-menthol and nimodipine [15]. In rabbit ventricular myocytes, these drugs inhibited the peak Ca$^{2+}$ current evoked by a step depolarization less potently than the late Ca$^{2+}$ current that remained at the end of the step depolarization, with nimodipine showing 13.1X difference in the IC$_{50}$s. The difference in IC$_{50}$s obtained for the step and the ramp current are compatible with literature findings of drugs exhibiting state- and/or use-dependent block of Ca$_V$1.2 channels in overexpression cell lines and L-type Ca$^{2+}$ channels in native myocytes. Nitrendipine [13, 17], nisoldipine, nicardipine [17], nifedipine, verapamil [16], and mibefradil [14] all showed more block when the cells were held at depolarized membrane potential than when cells were held at hyperpolarized membrane potential, suggesting preferential block of channels in the inactivated state. Verapamil and diltiazem showed block increasing at higher stimulation frequencies and higher depolarizations, suggesting a preference for the open and inactivated state over closed state [9–11, 31].

Figs 5 and 6 also show data summarizing inhibition of the Ba$^{2+}$ current by verapamil, buprenorphine, norbuprenorphine, naloxone, and diltiazem. All drugs showed greater inhibition of I$_{Ba-ramp}$ than I$_{Ba-step}$ (Figs 5 and 6) The largest difference was seen for verapamil, for which IC$_{50}$s between the step and the ramp current differed by a factor of 5. In external Ba$^{2+}$, the differences between the IC$_{50}$s of the step and the ramp current were more pronounced than in external Ca$^{2+}$ for verapamil, buprenorphine, norbuprenorpine, and naloxone. Even norbuprenorphine, which showed no difference between the IC$_{50}$s of I$_{Ca-step}$ and I$_{Ca-ramp}$, showed a difference of 2.4X between I$_{Ba-step}$ and I$_{Ba-ramp}$. These results thus demonstrate that state-dependent interactions between drugs and Ca$_V$1.2 channels, as well as relative preference for different channel states, are dependent on the charge carrier used. Generalizations regarding drug-Ca$_V$1.2 channel interactions based on different studies should thus take the charge carrier used into consideration.

Comparisons of the IC$_{50}$s on the same current region obtained in external Ba$^{2+}$ and Ca$^{2+}$ showed a small impact on the step current. Based on point estimate comparison, diltiazem showed the biggest difference, with an IC$_{50}$ at I$_{Ca-step}$ 2.1X higher than that at I$_{Ba-step}$. Likewise, the impact of charge carrier on the ramp current and the direction of shift were also drug-

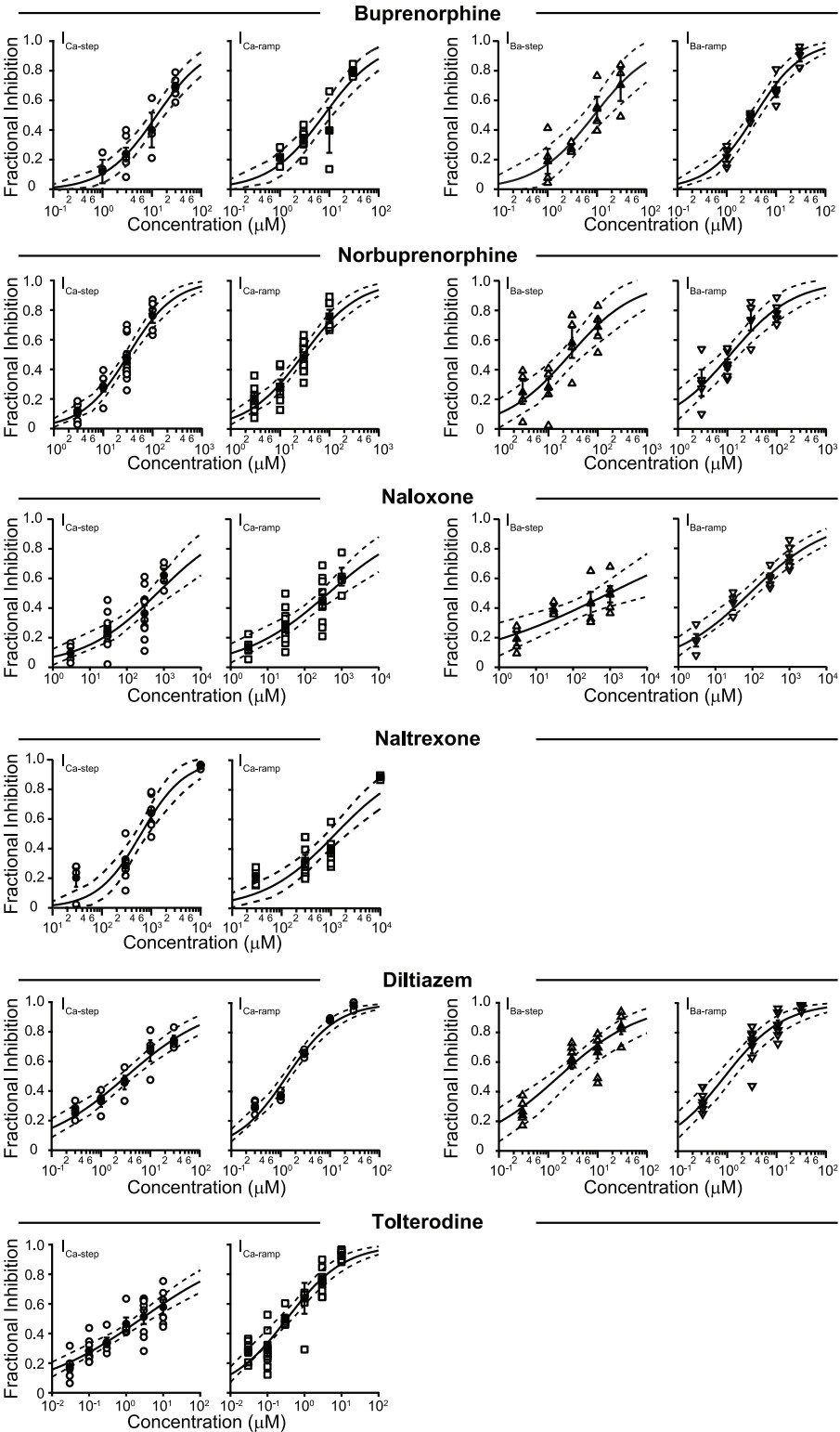

**Fig 5. Concentration-inhibition plots for Ca$_V$1.2 channel block by buprenorphine, norbuprenorphine, naloxone, naltrexone, diltiazem, and tolterodine at near PT.** Data for I$_{Ca-step}$ are shown in circles; I$_{Ca-ramp}$, squares; I$_{Ba-step}$, upright triangles; I$_{Ba-ramp}$, inverted triangles. Open symbols reflect individual data points; filled symbols plus error bars, mean ± sem. The solid sigmoidal curve indicates the fit with the Hill equation; the dashed curves, upper and lower limit of the 95% CI of the fit.

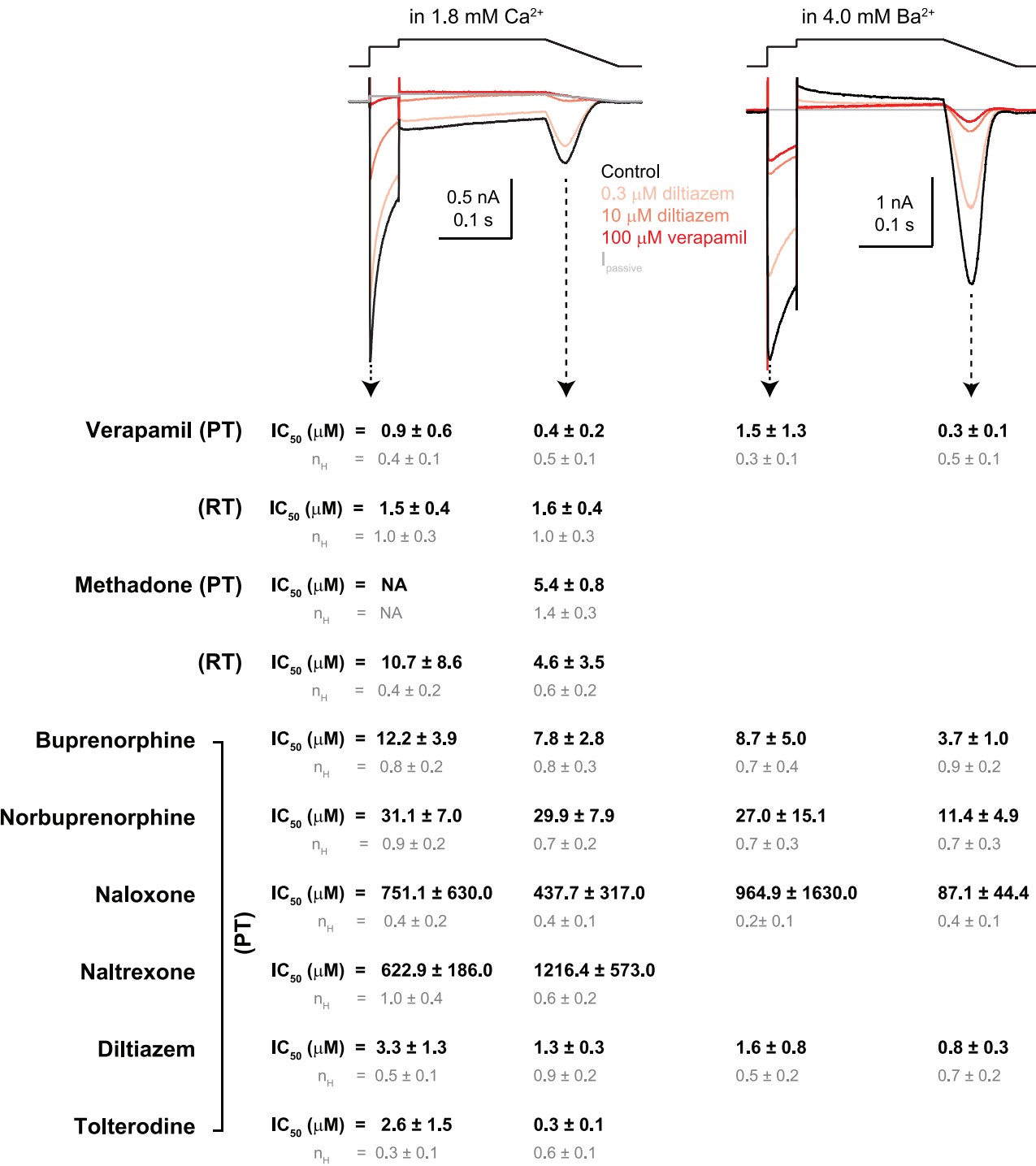

**Fig 6. Summary of IC$_{50}$ and n$_H$ values.** *Top*. The voltage protocol used and example traces recorded at near PT from two cells, one recorded in external Ca$^{2+}$ (*left*; cell ID: 21515003c) and the other in external Ba$^{2+}$ (*right*; cell ID: 21617012). For the cell on the left, the illustrated traces were obtained in the control solution (black; trace 206), 0.3 μM diltiazem (light red; trace 278), 10 μM diltiazem (medium red; trace 339), and 100 μM verapamil solutions (dark red; trace 397). For the cell on the right, the illustrated traces were obtained in the control solution (black; trace 50), 0.3 μM diltiazem (light red; trace 130), 10 μM diltiazem (medium red; trace 240), and 100 μM verapamil solutions (dark red; trace 286). I$_{passive}$ traces, shown in gray, were calculated based on R$_{input}$ derived from the verapamil traces shown. The voltage protocol was overlaid on top of the current traces. Note that for the cell on the right, Ba$^{2+}$ current isolation was done using I$_{passive}$ subtraction, as the outward current seen in control solution was no longer apparent in diltiazem and verapamil solutions. *Bottom*, summary IC$_{50}$ and n$_H$ values, (mean ± 95% CI) obtained at the current regions indicated by the dotted arrows.

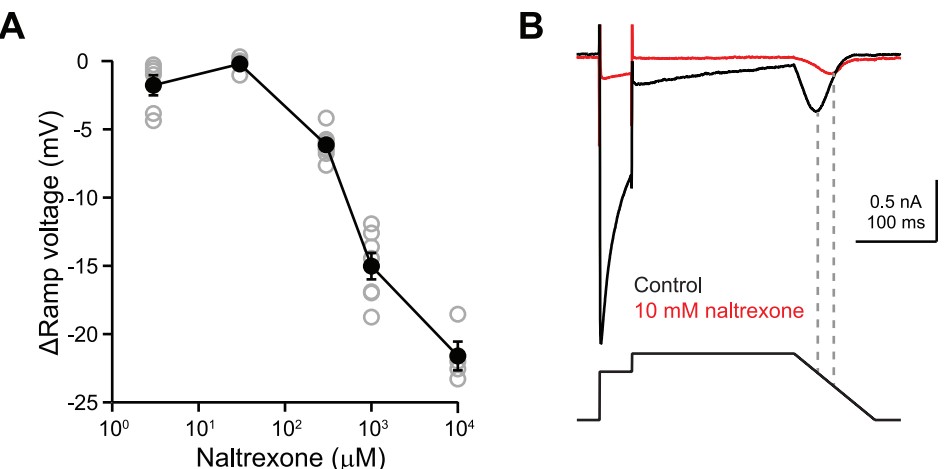

**Fig 7. Concentration-dependent shift of voltage at which I$_{Ca-ramp}$ peaked. A)** Changes in the ramp voltage vs. naltrexone concentrations. Gray open symbols indicated data points from individual cells; black filled symbols plus error bars indicate mean ± sem. **B)** Representative current traces obtained from 1 cell (cell ID: 20109006) in external Ca$^{2+}$ at near PT. The 90$^{th}$ trace (black) was the last trace obtained in the control solution; the 140$^{th}$ trace (red) was obtained following 10 mM naltrexone application. The voltage protocol was shown below the current traces. In this cell, the ramp voltage that I$_{Ca-ramp}$ reached peak shifted from –2 mV in the control solution to -18 mV following naltrexone application. For the naltrexone data set, the ramp voltage at which peak I$_{Ca-ramp}$ occurred in the control solution was 0.15 ± 0.35 mV (n = 21), consistent with the cells used to generate Fig 1C, 1D and 1G.

specific. Verapamil showed no difference, while naloxone showed a 5X difference, with I$_{Ba-ramp}$ being more sensitive to block than I$_{Ca-ramp}$. Results of the present study thus add to those in the literature demonstrating an impact of charge carrier on drug inhibition of Ca$^{2+}$ and Ba$^{2+}$ currents. In ventricular myocytes, diltiazem [11], D600 (a methoxy derivative of verapamil [11]), and verapamil [12] were less effective in inhibiting Ba$^{2+}$ current than Ca$^{2+}$ current through Ca$^{2+}$ channels [11]. Similar findings were reported for verapamil [10] and diltiazem [9] studied using Ca$_V$1.2 channels with β$_{1b}$ and α$_2$δ subunits using overexpression cells.

Data in Figs 5 and 6 also show that the n$_H$ values were quite variable amongst the drugs, ranging from very low (n$_H$ = 0.2 for naloxone on I$_{Ba-step}$) to quite steep (n$_H$ = 1.4 for methadone on I$_{Ca-ramp}$). Shallow slopes of the concentration-inhibition relationship may be reflective of technical challenges in the experiments. These challenges include current rundown that can inflate estimation of fractional inhibition at low drug concentration, and insolubility at high drug concentration that leads to fewer than expected free drug molecules to block ion channels. While these possibilities cannot be ruled out for tolterodine, they cannot explain data of other drugs. For verapamil and methadone, changing the recording temperature greatly altered the steepness of the concentration-inhibition curves, with n$_H$ values at one temperature being 0.4 to 0.6 and at the other temperature being 1 and above. For naltrexone and diltiazem, n$_H$ values differ between the step and the ramp current, with that for one current approaching 1 while the other remaining at or below 0.6. For naloxone and naltrexone, this laboratory has previous tested similar concentrations on Na$_V$1.5 channels and obtained n$_H$ values equal to or greater than 0.8 on the peak and the late current [18]. Therefore, the simplest explanation is that for these drugs, the shallowness of the concentration-inhibition relationship reflects more complex drug-Ca$_V$1.2 channel interactions that cannot be readily explained by 1:1 drug-receptor binding scheme that leads to immediate current inhibition. It is difficult to relate these n$_H$ results to existing literature, since n$_H$ values for concentration-inhibition plots are often not reported. In a study that assessed the structural basis of diltiazem block of voltage-gated Ca$^{2+}$ channels, the resting state block was found to have an IC$_{50}$ of 41 μM and a much steeper slope

for the concentration-inhibition relationship than use-dependent block, which had a shallower slope but more potent block with an IC$_{50}$ of 10.4 μM [32]. These results therefore demonstrate that n$_H$ values for diltiazem are state-dependent, consistent with the present findings. Based on X-ray crystallographic analysis, the study showed that diltiazem has two distinct binding poses with the Ca$^{2+}$ channels: upon entering the channel pore, this drug forms a loose channel-blocking complex that appears to be a low affinity binding mode, and then rearranges within the channel to a tighter binding, more stably blocking complex with diltiazem projecting into the selectivity filter from the central cavity upon voltage-dependent inactivation [32]. Importantly, diltiazem binding also allosterically modulates Ca$^{2+}$ binding in the selectivity filter, suggesting this mechanism may also contribute to reduction of current. Therefore, for a "pore blocker" like diltiazem, the different binding poses associated with different channel states may explain different n$_H$ values.

Table 2 summarizes the IC$_{50}$ values reported in the literature for methadone, diltiazem, verapamil, and tolterodine as well as experimental protocol used. A wide range of IC$_{50}$s were reported, with the max-to-min ratios for diltiazem being 397X (0.63 μM vs. 250 μM) and for verapamil 294X (0.16 μM vs. 47 μM). The only study that reported Ca$_V$1.2 channel block by buprenorphine, norbuprenorphine, naltrexone, and naloxone was by this laboratory [18]. In Tran et al., 2020, the IC$_{50}$ values of drug block on the ramp current measured using Ca$^{2+}$ as the charge carrier was derived from the same cells used in the present study.

## Limitation, lessons learned, protocol standardization, and conclusion

There are several limitations in the present study that can impact drug potency estimation. The first is that the drug concentrations exposed to the recorded cell were not measured using an analytical method. Drug concentrations can deviate from the target concentration due to compound-specific factors and human errors. The former include nonspecific binding to the plastic and glass substrates within the patch clamp perfusion apparatus, potential insolubility at higher concentrations tested, and instability in the perfusion solution under the experimental condition. Notably, verapamil and diltiazem are compounds stated to be light sensitive (https://www.sigmaaldrich.com/US/en/sds/sigma/v4629) and advised to keep away from direct sunlight (https://documents.tocris.com/pdfs/tocris_msds/0685_sds.pdf?1646647276&_ga=2.29634103.924187291.1646647248-600272910.1596464753), respectively, on the Safety Data Sheet from their distributors. While verapamil solutions and stocks were protected from light in this laboratory, diltiazem was not, and time-dependent degradation for these as well as other compounds tested throughout the recording day cannot be ruled out. Human errors can also occur during drug stock and drug solution preparation. Concentration verification, if possible, should be included as a part of the study design to rule out the possibility that deviations of drug concentrations translate into variability in drug potency estimation. The second limitation is that Ca$_V$1.2 current rundown under whole cell configuration could not be prevented, and current rundown was not corrected for drug potency estimation in this study. Although drugs were applied after the initial fast phase of rundown had ended, the IC$_{50}$s may have still been underestimated. Given cell-specific rundown profile, if rundown correction were to be implemented, then accounting for individual cell's rundown time course by fitting as many data points obtained in the control solution as possible, rather than using the time course plots derived from separate cells is recommended. Of note, although recordings in perforated patch configuration decrease cell dialysis-dependent rundown, this technique leads to higher R$_s$ compared to whole cell recordings. Current rundown was a trade-off for voltage control in the present study, and the method to optimize voltage control was to use whole cell recording to obtain as small of R$_s$ as possible followed by high degree of compensation. The third limitation

**Table 2. Comparisons of IC$_{50}$ values for Ca$_V$1.2 channel block for methadone, diltiazem, and verapamil generated by different experimental protocols.**

| Drug | Cell type | Subunits | Platform | Recording temperature (ºC) | Charge carrier (mM) | Voltage protocol | IC$_{50}$ and n$_H$ (± SE) | Reference |
|---|---|---|---|---|---|---|---|---|
| Methadone | CHO | hCa$_V$1.2, β$_2$, α$_2$δ | Automated (PatchXpress 7000A) | Ambient | NA | 2-pulse protocol: a 500 ms step from -80 to +10 mV, followed by repolarization to -80 mV for 500 ms, followed by a 200 ms step to +10 mV; the pulse pattern was repeated at 0.1 Hz | 26.3 μM (tonic block) 7.7 μM (phasic block) | [27] |
| Diltiazem | tsA-201 cells | Ca$_V$1.2, α$_2$δ, β$_{1b}$ | Manual | RT | 10 Ba$^{2+}$ | 100 ms pulses from -60 mV to +10 mV at 0.05 Hz | 65 ± 13 μM | [9] |
| | tsA-201 cells | Ca$_V$1.2, α$_2$δ, β$_{2a}$ | Manual | 22–25ºC | 20 Ba$^{2+}$ | 100 ms pulses from -80 mV to +20 mV at 0.2 Hz | 85 ± 9 μM | [38] |
| | HEK293 | Ca$_V$1.2, α$_2$δ, β$_{2a}$ along with K$_{ir}$2.3 | Automated (PatchXpress 7000A) and manual | Automated 22–26ºC; manual 20–25ºC | 30 Ba$^{2+}$ | 100 ms step from -60 to +20 mV at 0.05 Hz | 36 μM (automated) 44 μM (manual) | [39] |
| | Guinea pig ventricular myocytes | L-type Ca$^{2+}$ current | Manual | 20–25ºC | 10 Ca$^{2+}$ | 200 ms step from -70 to +10 mV at 0.2 Hz | 30 μM | [39] |
| | Guinea pig ventricular myocytes | L-type Ca$^{2+}$ current | Manual | RT, defined as 22–25ºC in [40] | 2 Ca$^{2+}$ | 200 ms step from -50 to +10 mV at 0.2 Hz | 0.63 ± 0.05 μM; n$_H$ = 0.77 | [41] |
| | CHO | hCa$_V$1.2, β$_2$, α$_2$δ | Automated (QPatch and PatchXpress) | Ambient | 1.8 Ca$^{2+}$ | 150 ms pulse from -40 mV to 0 mV every 5 s | 0.76 ± 0.08 μM; n$_H$ = 1.14 ± 0.15 | [33] |
| | *Trichoplusia ni* cells (insect) | Ca$_V$Ab (voltage-dependent Ca$^{2+}$-selective channel) | Manual | NA | 13 Ba$^{2+}$ | 20 ms single (resting block) vs 20 pulses at 1 Hz (use-dependent block) from -120 mV; amplitude unknown | Resting state block: 41 μM Use-dependent block: 10.4 μM | [32] |
| | HEK293 | Ca$_V$1.2, α$_2$δ, β$_{2a}$ | Manual | ~25ºC | 1.8 Ca$^{2+}$ 5 mM Ba$^{2+}$ | 100 ms step from -90 to 10 mV at 0.03 Hz | 250 ± 16 μM (Ca$^{2+}$) 109 ± 12 μM (Ba$^{2+}$) | [42] |
| Verapamil | Guinea pig ventricular myocytes | L-type Ca$^{2+}$ current | Manual | RT, defined as 22–25ºC in | 2 Ca$^{2+}$ | 200 ms step from -50 to +10 mV at 0.2 Hz | 0.16 ± 0.01 μM; n$_H$ = 0.61 | [41] |
| | hCav1.2 from Chantest | NA | IonWork Quattro | RT | NA | From -65 mV (applied for 10 s) a 500 ms depolarizing step to 0 mV was applied | 2.69 μM, converted from pIC$_{50}$ of 5.571 | [43]; experimental details in [44] |
| | *Trichoplusia ni* cells (insect) | Ca$_V$Ab (voltage-dependent Ca$^{2+}$-selective channel) | Manual | NA | 13 Ba$^{2+}$ | 20 pulses at 1 Hz | 475 ± 25 nM | [45] |
| | HEK293 | Ca$_V$1.2, α$_2$δ, β$_{2a}$ along with K$_{ir}$2.3 | Automated (PatchXpress 7000A) and manual | Automated 22–26ºC; manual 20–25ºC | 30 Ba$^{2+}$ | 100 ms step from -60 to +20 mV at 0.05 Hz | 15 μM (automated) 47 μM (manual) | [39] |
| | Guinea pig ventricular myocytes | L-type Ca$^{2+}$ current | Manual | 20–25ºC | 10 Ca$^{2+}$ | 200 ms step from -70 to +10 mV at 0.2 Hz | 0.79 μM | [39] |
| | CHO | hCa$_V$1.2, β$_2$, α$_2$δ | Automated (QPatch and PatchXpress) | Ambient | 1.8 Ca$^{2+}$ | 150 ms pulse from -40 mV to 0 mV every 5 s | 0.20 ± 0.02 μM; n$_H$ = 0.80 ± 0.11 | [33] |
| | CHO | Cav1.2, β$_2$, α$_2$δ$_1$ | IonWorks Barracuda | Ambient | 7 Ca$^{2+}$ | 2-pulse protocol: from -80 mV two 250 ms steps to +10 mV separated by 1s in between; Pre-conditioning protocol: from -80 mV step to -40 mV for 50 s then to +10 mV for 250 ms | Two-pulse protocol: Test pulse 1: 34.9 μM Test pulse 2: 9.8 μM Pre-conditioning protocol: 1.89 μM | [16] |
| | tsA-201 cells | Ca$_V$1.2, α$_2$δ, β$_{1b}$ | Manual | RT | 10 Ba$^{2+}$ | 100 ms pulses from -60 mV to +10 mV at 0.05 Hz | 40.5 ± 0.9 μM | [10] |
| | HEK Ca$_V$1.2 | NA | Manual | Near physiological temperature | 1.8 Ca$^{2+}$ | From -80 mV step to 0 mV for 40 ms then to +30 mV for 200 ms then ramp down to -80 mV within 100 ms (1.2 V/s); protocol was repeated at 0.2 Hz | 0.206 μM* | [46] |

*It is unclear whether the IC$_{50}$ was measured at the 0 mV step or the ramp down phase in this study.

is that R$_s$ was not measured throughout the recordings. Not having this measure across time raises a logical concern that rundown of Ba$^{2+}$ and Ca$^{2+}$ current is due to large R$_s$ changes. In this study, Ca$^{2+}$ and Ba$^{2+}$ currents recorded at near PT activate extremely fast, and large changes in R$_s$ manifest as slowing of time-to-peak for the step current (for Ba$^{2+}$ and Ca$^{2+}$

currents) and shift in the voltage at which ramp peak occurred (for the Ba$^{2+}$ current; Fig 3). Therefore, whether R$_s$ changed dramatically or not during the control recording period was based on offline assessment of these current profiles. Progressive decreases in the current amplitude not accompanied by kinetic changes seem incompatible with the conjecture that rundown is secondary to large changes in R$_s$. As the original electrophysiology records are available at https://osf.io/g3msb/, interested readers are encouraged to assess these files to draw independent conclusions regarding the mechanisms subserving rundown. The fourth limitation is that the reversal potentials of Ca$^{2+}$ and Ba$^{2+}$ currents were extrapolated from currents obtained between 0 and +20 mV steps (S5 Fig). These I-V relations were generated to assess adequacy of voltage control, inferred from graded increases in the current amplitudes to increasing voltages between -60 to -20 mV. For the purpose of measuring the reversal potential, extending the voltage steps to beyond the reversal potential would provide direct measurement for each cell. The fifth limitation is the uncertainty that the recorded Ca$_V$1.2 current indeed reflects activity of channels with $\beta_2$ and $\alpha_2\delta_1$ auxiliary subunits. The gray traces in Fig 1G and 1H show that the ratios of ramp-to-step current are quite variable across cells. Likewise, temperature sensitivity of the whole cell current was also quite different across cells (Fig 2E and 2J). Since auxiliary subunits modulate Ca$_V$1.2 channel gating, it is possible that recordings in this study were from heterogenous Ca$_V$1.2 channels with either one or both auxiliary subunits or channels in different states of phosphorylation.

A few lessons were learned by conducting the present study. First, data variability of Ca$_V$1.2 channel block was collectively larger than those observed for cardiac hERG and Na$_V$1.5 channel block based on this laboratory's concurrent work. The concentration-inhibition plots presented in Figs 4 and 5 show variable degrees of current inhibition for individual cells to a given drug concentration, with no clear outliers observed. This level of data spread was not seen for hERG and Na$_V$1.5 current inhibition by the same drugs [18]. Within-the-study data variability for Ca$_V$1.2 current inhibition may be due to variable degrees of current rundown and heterogenous coupling between the channel $\alpha$ subunit with $\beta_2$ and $\alpha_2\delta_1$ subunits, as the latter can also be targets of drugs. Second, state-dependent block of drugs on Ca$_V$1.2 channels are common, inferred from the different IC$_{50}$s obtained for the step and the ramp current. Understanding state-dependent block using voltage protocols that recapitulate cardiac AP may be important when trying to predict drug impact on cardiac electrophysiology. In the sinoatrial nodal cells, Ca$_V$1.2 channels are activated rapidly upon membrane depolarization and contribute to the upstroke of the AP. Drug block of open Ca$_V$1.2 channels (i.e., inhibition of the step current) may thus inform the potential of drug in affecting the heart rate and the PR interval. On the other hand, during a ventricular AP, Ca$_V$1.2 channels are activated by the abrupt depolarization from rest, enter inactivated state during the plateau potential, and then become reactivated during the repolarizing phase of the AP before entering closed state. Information regarding how Ca$_V$1.2 channel block develops during a ventricular AP, as well as drug effect on voltage-dependence of channel gating may inform potential change in AP shape (i.e., triangulation or simply shortening), thereby allowing better prediction of proarrhythmia risk. Some statistical [33] and *in silico* myocyte models have incorporated drug block of Ca$_V$1.2 channels to assess the risk of *Torsade de Pointes* imposed by hERG channel block [34]. Incorporating IC$_{50}$s measured at distinct current regions or accounting for state-dependent block of Ca$_V$1.2 channels may lead to better model performance. The third lessons learned is that many drugs have n$_H$ values much smaller than 1, and these values are dependent on the recording temperature and channel state (Fig 6). Once the contributions of current rundown and insolubility at higher tested concentrations to shallow concentration-inhibition graphs are ruled out, the most straightforward interpretation of these results is that drug-Ca$_V$1.2 channel interactions leading to current inhibition are complex processes that may involve multiple binding

poses (i.e., diltiazem), multiple binding sites (i.e., nitrendipine, roscovitine), or through allosteric mechanisms (i.e., diltiazem; naltrexone, Fig 7).

The present results showed that even with the same voltage protocol presented at the same stimulation rate, Cav1.2 pharmacology can still be sensitive to a variety of factors encountered during the experiments and during data analysis. When numerous experimental factors are different between two studies, as seen Crumb et al., 2016 [5] and Li et al., 2018 [6], drug potency estimates can be very different even for the same drugs. For Ca$_V$1.2 data intended to support risk prediction or clinical interpretation, normalizing laboratory-specific practices is essential toward promoting data reproducibility across laboratories—a pivotal step toward engendering confidence amongst regulators for applying these *in vitro* data in the decision-making process. Toward this end, the FDA Cardiac Safety Studies Interdisciplinary Review Team (CSS-IRT) has posted a document regarding the recommended voltage protocols for cardiac ion channels, including Ca$_V$1.2 channels, on its website (https://www.fda.gov/media/151418/download). The voltage waveform, stimulation frequency, compositions of the internal and Ca$^{2+}$-based external solutions, and data analysis method are consistent with those used in the present study. For cardiac safety assessment, drug developers and regulators are following the guidelines released by the International Council for Harmonisation for Technical Requirements for Pharmaceuticals for Human Use: ICH S7B for nonclinical [35] and ICH E14 for clinical studies [36]. The newly released ICH E14/S7B Questions and Answers guideline offers best practice recommendations for patch clamp ion channel studies intended to support cardiac safety assessment [37] (ICH S7B Q&A 2.1), and the protocol used in this manuscript is consistent with these recommendations. ICH S7B Q&A 2.1 also recommends recording at near PT. This study tested two temperature controller models. Given the gravity-fed perfusion method and shallow bath chambers used here, the temperature controller model that heats the chamber bottom uniformly provided more stable temperature control. However, if a perfusion pump were used to maintain flow rate, then conceivably the temperature controller model that maintained bath temperature by heating the anodized aluminum platform would have also achieved stable temperature control. Experimenters interested in measuring drug block of Ca$_V$1.2 channels at near PT are recommended to consider how bath temperature may fluctuate given the rig design, and importantly measure bath temperature near the recorded cell throughout the recordings to enable subsequent analysis of temperature fluctuations on within-the-study data variability. In conclusion, results from the present study offer rationale for the best practice recommendations regarding experimental design, conduct, and data quality consideration, and may benefit stakeholders considering utilizing Ca$_V$1.2 channel data to support regulatory decision-making.

This table summarizes the IC$_{50}$ and n$_H$ values for Ca$_V$1.2 channel block determined using the manual patch clamp method [5] and automated patch clamp system [6]. While these studies examined more overlapping drugs, only those for which Crumb et al., 2016 provided IC$_{50}$ and n$_H$ values are shown for comparison. The ratios are calculated as maximum vs. minimum IC$_{50}$s. Crumb et al., 2016 used a Ca$_V$1.2-CHO cell line from Cytocentrics Bioscience GmbH (Rostock, Germany), and did not provide information regarding subunits expressed. Experiments were conducted using whole cell patch clamp method at 36 ± 1ºC, and current was evoked using a rabbit ventricular AP waveform repeated at 10 s interval. Ba$^{2+}$ (4 mM) was used as the charge carrier. The automated patch clamp data from Li et al., 2018 were generated using a CHO cell line that expressed hCa$_V$1.2α, β$_2$, α$_2$δ$_1$ subunits from Charles River Laboratories (Wilmington, MA). Recordings were performed using IonWorks Barracuda system operating in population perforated patch clamp mode. Recording temperature was not controlled and was expected to be higher than RT due to the heat produced during system operation. Inward current was evoked using the same "step-step-ramp" voltage waveform as used in the

present study but repeated at 10 s interval. The current elicited by the 0 mV step was used to quantify drug effects. Ca$^{2+}$ (6.8 mM) was used as the charge carrier. Compositions of the external recording solution were the same for both studies except for the charge carrier. For internal solution, Crumb et al., 2016 used 130 mM CsCl as the main salt, while Li et al., 2018 used 90 mM CsF + 50 mM CsCl. Current mediated by Ca$_V$1.2 channels exhibit prominent rundown when recorded under the whole cell configuration. Percent current inhibition by drug reported by Li et al. 2018 was adjusted for current run down, using data from vehicle and positive control wells, even though recordings were obtained using perforated patch mode. Crumb et al., 2016 did not correct for current rundown nor specified the rate of rundown for the cells used. The predicted logP values for these drugs based on ChemAxon are provided as estimates of lipophilicity. The sources are as follows: bepridil (https://go.drugbank.com/drugs/DB01244), chlorpromazine (https://go.drugbank.com/drugs/DB00477), diltiazem (https://go.drugbank.com/drugs/DB00343), ondansetron (https://go.drugbank.com/drugs/DB00904), terfenadine (https://go.drugbank.com/drugs/DB00342), and verapamil (https://go.drugbank.com/drugs/DB00661). There is no relationship between the ratio of max-to-min IC$_{50}$s and logP values for these drugs.

## Supporting information

**S1 Fig. I$_{passive}$-subtraction vs. verapamil-subtraction.** Recordings were obtained in external Ca$^{2+}$. Cell ID was 18n20001 for panels **(A)** through **(C)**; 18n14007, panels **(D)** through **(I)**. Dashed lines in panels **(A)**, **(B)**, **(D)**, **(F)**, and **(H)** mark the 0 pA level. **A)** Original unsubtracted traces from a cell for which I$_{passive}$ subtraction method worked well to quantify I$_{Ca-ramp}$. Traces 1 and 35 were the 1$^{st}$ and last recorded traces in control solution (black). Trace 85 was the last trace recorded in 30 nM tolterodine (light red). Trace 126 was the last trace recorded in 100 μM verapamil (red). I$_{passive}$, calculated using R$_{input}$ derived from trace 126, is shown in gray. Note the good alignment between I$_{passive}$ and trace 126 at all voltages, suggestive of little to no endogenous voltage-dependent current in this cell under the specified experimental condition. **B)** I$_{passive}$-subtracted traces show little to no outward current that is above 0 pA. I$_{Ca-ramp}$ for this cell is quantified as the peak inward current during the voltage ramp down phase using I$_{passive}$-subtracted traces. **C)** Time course plots of I$_{Ca-ramp}$ (*top panel*), R$_{input}$ (*middle panel*), and I$_{-80 \text{ mV}}$ (*lower panel*) for the same cell. **D)** Original unsubtracted traces from a cell for which verapamil subtraction method was used to quantify I$_{Ca-ramp}$. Traces 1 and 62 were the 1$^{st}$ and last recorded traces in control solution (black). Trace 114 was the last trace recorded in 3 μM tolerodine (medium red). Trace 143 was the last trace recorded in 100 μM verapamil (red). Note that this cell had larger outward current at the +30 mV step and the voltage ramp down phase than the cell illustrated above that was unmasked when the inward current was reduced (see medium red and red traces). **E)** *Top*, time course plots of absolute current amplitude, measured with a 20 ms window around the inward current peak (corresponding to ramp voltage 10.8 to -39.5 mV). Given the presence of outward current, as inward Ca$^{2+}$ current was suppressed with verapamil, polarity of the absolute current reversed from being negative to positive. The middle and lower panels are time course plots of R$_{input}$ and I$_{-80 \text{ mV}}$ for this cell. **F)** I$_{passive}$-subtracted traces for the same cell showed that outward current remained. **G)** Voltages at which the maximal negative ramp current was identified from I$_{passive}$-subtracted traces and by searching the entire voltage ramp down phase. Similar to panel **(E)**, there was a jump of voltage at which the maximal negative ramp current was identified when Ca$^{2+}$ current was largely inhibited (i.e., ramp current approaching linear). **H, I)** Verapamil-subtracted traces for this cell **(H)** and time course plot of I$_{Ca-ramp}$ quantified using verapamil-subtracted traces **(I)**. (EPS)

**S2 Fig. Rundown of Ca$^{2+}$ current for two additional cell lines. A, B)** These panels show Ca$^{2+}$ currents recorded from 2 additional cell lines. Currents were also mediated by hCa$_v$1.2α, β$_2$, and α$_2$δ$_1$ subunits expressed in HEK293 cells. Verapamil was not applied for cell in **(B)** since there was little Ca$_V$1.2 current remaining after 150 traces. **C, D)** Summary time course plots of I$_{Ca-step}$ **(C)** and I$_{Ca-ramp}$ **(D)** recorded in control solution for the cell line represented by **(A)**. **E, F)** Summary time course plots of I$_{Ca-step}$ **(E)** and I$_{Ca-ramp}$ **(F)** for the cell line represented by **(B)**. Note that some cells did not last for all 200 traces of recording. Verapamil was not applied for this cell line.
(EPS)

**S3 Fig. Time course plots demonstrating amplitude fluctuations for I$_{Ca-0\ mV}$ but not I$_{Ca-ramp}$.** Recordings occurred in the control solution followed by bath application of 100 μM verapamil. Cell ID was 19322002. **A)** *Top*, current traces 100 and 114 (black) were obtained in the control solution; trace 200 (red), following steady state current inhibition by verapamil. I$_{passive}$, shown in gray, was derived from the verapamil trace. *Bottom*, the voltage protocol used. **B)** The time course plots of I$_{Ca-0\ mV}$ and I$_{Ca-ramp}$. **C)** R$_{input}$ (*top*) and I$_{-80\ mV}$ (*bottom*).
(EPS)

**S4 Fig. Current-temperature relationship for additional cells shown in Fig 2E and 2J.** Cells shown in panels **(A)** through **(E)** were recorded in 1.8 mM Ca$^{2+}$; panels **(F)** through **(I)**, in 4.0 mM Ba$^{2+}$. For each panel, the cell ID is shown on the top of the plots. **A)** This cell had 3 cycles of temperature manipulations and had rundown correction performed for both I$_{Ca-0\ mV}$ and I$_{Ca-ramp}$. Fitting the rundown-corrected I$_{Ca-step}$-temperature relation with a linear function yielded a slope of -0.0405 nA/°C and a Y-intercept of 0.704 nA. No relation was observed between I$_{Ca-ramp}$ and temperature. **B)** This cell had 5 cycles of temperature manipulations and did not require rundown correction of the current amplitudes. The slope of the I$_{Ca-0\ mV}$-temperature relation was -0.121 nA/°C, and the Y-intercept was 3.406 nA. No relation was observed between I$_{Ca-ramp}$ and temperature. **C)** This cell had 5.5 cycles of temperature manipulations and did not require of rundown correction of the current amplitudes. The slope of the I$_{Ca-step}$-temperature relation was -0.0278 nA/°C, and the Y-intercept was 0.342 nA. I$_{Ca-ramp}$ was also inversely related to temperature for this cell. The slope of the I$_{Ca-ramp}$-temperature relation was -0.0160 nA/°C, and the Y-intercept was 0.0694 nA. **D)** This cell had 2 cycles of temperature manipulations and did not require rundown correction of current amplitudes. The slope of the I$_{Ca-step}$-temperature relation was -0.129 nA/°C, and the Y-intercept was 2.66 nA. No relation was observed between I$_{Ca-ramp}$ and temperature (r = 0.27). **E)** This cell had 2 cycles of temperature manipulations and did not require rundown correction of current amplitudes. The slope of I$_{Ca-0\ mV}$-temperature relation was -0.288 nA/°C, and the Y-intercept was 8.059 nA. I$_{Ca-ramp}$ was also inversely related to the bath temperature for this cell. The slope of the I$_{Ca-ramp}$-temperature relation was -0.0301 nA/°C, and the Y-intercept was 0.808 nA. **F)** This cell had 4 cycles of temperature manipulations. Rundown correction was performed for both I$_{Ba-step}$ and I$_{Ba-ramp}$. The slope of rundown corrected I$_{Ba-step}$-temperature relation was -0.117 nA/°C, and the Y-intercept was 1.414 nA. I$_{Ba-ramp}$ appeared to be inversely related to the bath temperature. The slope of rundown corrected I$_{Ba-ramp}$-temperature relation was -0.0292 nA/°C, and the Y-intercept was -0.771 nA. **G)** This cell had two cycles of temperature manipulations. Rundown correction was performed on both I$_{Ba-0\ mV}$ and I$_{Ba-ramp}$. The slope of the I$_{Ba-0\ mV}$-temperature relation was -0.146 nA/°C, and the Y-intercept was 2.360 nA. I$_{Ba-ramp}$ appeared to be inversely related to the bath temperature. The slope of I$_{Ba-ramp}$-temperature relation was -0.0225nA/°C, and the Y-intercept was -0.558 nA. **H)** This cell had 5 cycles of temperature manipulations and did not require rundown correction of current amplitudes. The slope of the I$_{Ba-step}$-temperature relation was -0.120 nA/°C, and the Y-intercept was 3.165

nA. I$_{Ba-ramp}$ was also inversely related to the bath temperature. The slope of the linear fit was -0.0603 nA/˚C, and the Y-intercept was 1.426 nA. **I)** This cell had 5 cycles of temperature manipulations and did not require rundown correction of current amplitudes. The slope of I$_{Ba-step}$-temperature relation was -0.287 nA/˚C, and the Y-intercept was 7.098 nA. I$_{Ba-ramp}$ was also inversely related to the bath temperature. The slope of the I$_{Ba-ramp}$-temperature relation was -0.0770 nA/˚C, and the Y-intercept was 0.774 nA.
(EPS)

**S5 Fig. Normalized I-V relation of Ba$^{2+}$ and Ca$^{2+}$ current.** Currents were evoked from a holding potential of -80 mV to +20 mV in external Ba$^{2+}$ **(A)** or +15 mV in external Ca$^{2+}$ **(B)** with 5 ms voltage steps in 5 mV increments. For each cell, the peak current evoked by each voltage step was normalized to the maximum current recorded. The normalized and averaged current from all cells for a particular voltage step was then plotted against that membrane voltage to generate the normalized I-V plots. Reversal potentials (E$_{rev}$) of the Ba$^{2+}$ and Ca$^{2+}$ currents were estimated by fitting the averaged data points from -5 mV to +20 mV in external Ba$^{2+}$ **(A)** or -5 mV to +15 mV in external Ca$^{2+}$ **(B)** with linear functions in the form of y = a + bx, and then solving for x when y = 0. For **(A)**, a = 0.794; b = -0.023. For **(B)** a = 0.799; b = -0.017.
(EPS)

**S6 Fig. Exemplar time course plots of Ca$^{2+}$ current recorded at near PT following applications of: A) buprenorphine, B) norbuprenorphine, C) methadone, D) naltrexone, E) naloxone, and F) tolterodine.** Open black symbols reflect absolute current amplitudes obtained by analyzing unsubtracted current traces; open red symbols reflect I$_{passive}$-subtracted current amplitudes. All plots show absolute and I$_{passive}$-subtracted amplitudes. Due to current rundown, data points for earlier traces for some cells are off the scale hence not illustrated.
(EPS)

**S7 Fig. Exemplar time course plots of Ba$^{2+}$ current recorded at near PT following applications of: A) buprenorphine, B) norbuprenorphine, and C) naloxone.** Open black symbols reflect absolute current amplitudes obtained by analyzing unsubtracted current traces; open red symbols reflect I$_{passive}$-subtracted current amplitudes. All plots show absolute and I$_{passive}$-subtracted amplitudes. Due to current rundown, data points for earlier traces for these cells are off the scale hence not illustrated.
(EPS)

## Author Contributions

**Conceptualization:** Wendy W. Wu.

**Data curation:** Ming Ren, Aaron L. Randolph, Claudia Alvarez-Baron, Donglin Guo, Phu N. Tran, Nicolas Thiebaud, Jiansong Sheng, Jun Zhao, Wendy W. Wu.

**Formal analysis:** Ming Ren, Aaron L. Randolph, Claudia Alvarez-Baron, Donglin Guo, Nicolas Thiebaud, Jiansong Sheng, Jun Zhao, Wendy W. Wu.

**Investigation:** Ming Ren, Aaron L. Randolph, Claudia Alvarez-Baron, Donglin Guo, Phu N. Tran, Nicolas Thiebaud, Jiansong Sheng, Jun Zhao, Wendy W. Wu.

**Methodology:** Ming Ren, Aaron L. Randolph, Claudia Alvarez-Baron, Wendy W. Wu.

**Supervision:** Wendy W. Wu.

**Validation:** Ming Ren, Claudia Alvarez-Baron, Wendy W. Wu.

**Visualization:** Wendy W. Wu.

**Writing – original draft:** Aaron L. Randolph, Claudia Alvarez-Baron, Wendy W. Wu.

**Writing – review & editing:** Ming Ren, Aaron L. Randolph, Claudia Alvarez-Baron, Donglin Guo, Phu N. Tran, Nicolas Thiebaud, Jiansong Sheng, Jun Zhao, Wendy W. Wu.

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
