## [Decision Letter · Decision Letter 0]

1 May 2022

PONE-D-22-10448Experimental factors that impact CaV1.2 channel pharmacology – effects of recording temperature, charge carrier, and current region of analysisPLOS ONE

Dear Dr. Wu,

Thank you for submitting your manuscript to PLOS ONE. After careful consideration, we feel that it has merit but does not fully meet PLOS ONE’s publication criteria as it currently stands. Therefore, we invite you to submit a revised version of the manuscript that addresses the points raised during the review process. Your manuscript has been read and evaluated by 3 expert reviewers and a number of issues have been noted, for example, some more explanation of the rationale for the exact protocol used to generate ICaL as well as further information regarding series resistance (Rs) and the measured membrane resistance (Rm). All comments raised by the Reviewers should be addressed in a revised manuscript

We look forward to receiving your revised manuscript.

Kind regards,

Daniel M. Johnson, PhD

Academic Editor

PLOS ONE

Journal Requirements:

5. Please ensure that you refer to Supllemental Figure 1-4 in your text as, if accepted, production will need this reference to link the reader to the figure.

Reviewers' comments:

Reviewer's Responses to Questions

**Comments to the Author**

1. Is the manuscript technically sound, and do the data support the conclusions?

Reviewer #1: Yes

Reviewer #2: Yes

Reviewer #3: No

2. Has the statistical analysis been performed appropriately and rigorously? 

Reviewer #1: I Don't Know

Reviewer #2: Yes

Reviewer #3: I Don't Know

3. Have the authors made all data underlying the findings in their manuscript fully available?

Reviewer #1: Yes

Reviewer #2: Yes

Reviewer #3: No

4. Is the manuscript presented in an intelligible fashion and written in standard English?

Reviewer #1: Yes

Reviewer #2: Yes

Reviewer #3: Yes

5. Review Comments to the Author

Reviewer #1: The study from Ren and colleagues performs a dissection of key experimental factors affecting the outcome of pharmacological studies on Cav1.2 currents in an expression system. Up to 8 drugs are studied.

----- Lack of novelty -----

The study is well written, detailed in its methodology, very transparent and it is very difficult to argue with in its conclusions. This said, in my personal opinion the main weakness of the study is that it lacks scientific novelty: it was already known that temperature, degree of Rs compensation, charge carrier and the voltage-clamp protocol used are factors that affect the outcome of experiments.

----- Recommendations at the end of the paper are missing -----

My most constructive suggestion to improve the manuscript’s value is as follows: In my opinion, the manuscript’s call to standardize conditions among laboratories is fully justified. However, this corollary is not accompanied by EXPLICIT unifying criteria or explicit protocolary recommendations proposed by the authors that could help other laboratories to standardize such conditions. In the absence of such explicit recommendations, just pointing to the problem (which was known, see "Lack of novelty" above) is not a solution… In summary: how does this study contribute to solving the problem?

While some of such recommendations are obvious (e.g. compensate Rs to 70-80%), others are not (e.g. how-to best control temperature to reduce variability? statistics?). Topics that come to my mind and that could be addressed explicitly include:

(1) All data considered… is it more advisable to perform a step protocol or a step-step-ramp protocol while testing a given compound, from the points of view of variability and accuracy? In other words, which protocol has the highest changes of being reproduced among labs with the lowest variability, while still obtaining the proper drug potency and IC50?

(2) Other protocols that should be considered? For example, in the study limitations the AP-clamp protocol is considered. What about 2-pulse protocols for recovery from refractoriness and/or inactivation properties?

(3) Any proposal to better control the temperature such as a given chamber of definite dimensions, flow rate, etc? (the apparatus that works best is already mentioned). Any material that works best to avoid drug binding to it?

(4) When working at room temperature, can suitable correction factors be found or, because of additional issues such as differential drug binding at different temperatures, there is no point? How does this vary among drugs, given their known mechanism(s) of action?

(5) When not addressing mechanistic studies (e.g. VDI vs. CDI in the presence of a drug), should results in the presence of calcium (as charge carrier) be preferred to those in barium because they reflect more faithfully the cellular physiology?

(6) How should data in expression systems be compared vs. data in, for example, ventricular myocytes or human iPSC-derived cardiomyocytes? It is important to acknowledge that myocytes experience a quantitatively strong calcium-dependent inactivation from SR calcium being released, as compared to expression systems where such SR release does not exist.

(7) Should laboratories provide WB data on the expression of subunits vs. a well-characterized, commercial cell line, available to all labs?

I have to add 2 extra recommendations, which are not very clearly stated in the methods section:

(8) Any recommendation on confidence intervals when performing a fit?

(9) Any recommendation about finding outliers, given the strong inter-cell variability? For example: should outliers be considered prior to drug administration, after computing an inhibition ratio, or both?

----- Readability -----

Perhaps for its amount of detail (which favors reproducibility), the paper does sometimes flow very slowly, and this could discourage readers. In my opinion, it may be best if authors transferred some of their written results to tables in order to summarize (so that readers remain focused).

----- Comparison with previous literature is performed qualitatively -----

I missed a table comparing the estimated IC50 values in this study with those values from prior literature. The 2 studies cited (refs. 5 and 6) are extreme examples for their differences, but there should be more studies to compare with. Where (within the known literature range) do the current IC50s dwell?

In fact, I missed a table or paragraph describing what was known about each of the tested drugs before this study, as a part of the introduction. Interested readers who are not experienced may get easily lost or fail to understand if the drug mechanisms proposed in this study have been already described or not.

----- Question about temperature dependency -----

In the traces shown in Figure 2B, the curve in ICa-ramp amplitude appears to be time-shifted (i.e. delayed) vs. the temperature curve, rather than being totally unrelated to temperature. Have the authors considered the possibility of such delay and whether it applies to cells where no current-temperature relationship was observed? A correlation with some temporal shift between the curves could be a good way to demonstrate (presence or absence) this behaviour

----- Info missing in methods -----

Please state lag (microseconds) when applying Rs compensation.

Please state R(input) and Rs tolerance limits (i.e. for R(input) and Rs to be considered constant during an experiment: for example “R(input) and Rs were considered constant if they did not change by more than XXXXX percent during the experiments”.

Please see topics (8) and (9) above and provide additional information about the fitting procedure and the identification of outliers

----- Minor -----

By reading the title, I could not understand the words “current region of analysis”. It took a while, until the authors described the “step-step-ramp” protocol, that the title made full sense to me. I would suggest rewording of that part.

Reviewer #2: Temperature dependence of ion channel pharmacology, particularly at physiological temperatures, is very important. However, despite this importance it is under investigated, so I applaud the authors for their thorough examination in Cav1.2 ion channels. The challenges, some of the solutions found & highlighting other areas that require further work will be useful to the field, & will help to get greater consistency & physiologically relevant comparisons between labs in critical areas like cardiac ion channel safety pharmacology.

I suggest the following Qs & areas should be addressed within the MS to help improve & explain the findings for the reader:

1. Line 29: “Three protocols were used for tetracycline induction.” Why three & was any one protocol preferred over others e.g. pros/cons that may explain the choice?

2. Line 48: “Two [temperature] controller models were used …” Nice explanation of the two controllers. But seems that in both intra-bath temperature variations & in overall temperature consistency that the TC2BIP was superior. Yet it seems despite this that the TC-344C was used (line 318: “Therefore, temperature sensitivity of the CaV1.2 current was examined. These experiments were conducted using setups with the TC-344C controller …”). Why?

3. Line 272: “The normalized ICa-step at the 150th trace was 0.42 ± 0.04 relative to the 1st trace (Fig. 1C); the normalized ICa-ramp, 0.44 ± 0.06 (Fig. 1D). The normalized IBa-step was 0.34 ± 0.04 (Fig. 1E); the normalized IBa-ramp, 0.32 ± 0.04 (Fig. 1F).” Are the bigger post-run-down currents for Ca2+ (~0.43 nA) vs Ba2+ (~0.33nA) significantly different? If so, does that suggest a Ca-dependent reduction in run-down? (It seems a little counter-intuitive, since Ba obviously reduces current Ca-dependent inactivation, as explained by the authors. However, such inactivation is temporary & mostly recovered between depolarising pulses, whilst run-down is pseudo-permanent & cumulative over the time-course of such experiments.)

4. Lines 595-606: V good description & explanation of possible solutions & analysis for run-down currents. Another possible issue that could be highlighted here is that the effects recorded after run-down might be vs a specific channel population, such as specific subunit composition(s) &/or phosphorylation state(s) mentioned in the third limitation (lines 607-613) i.e. waiting to get a plateau current after run-down may mean you’ve skewed your recordings to a subpopulation of channels &/or phosphorylation state.

5. Line 643: “… multiple binding poses (i.e., diltiazem), multiple binding sites (i.e., nitrendipine, roscovitine), or through even allosteric mechanisms (i.e., diltiazem; naltrexone, Fig. 7).” Even needs deleting.

6. Line 659: Table 1 – it might be worth adding an extra column to this table to give logP (measure of lipophilicity) of these compounds. It would be interesting to see if these ratios correlate to the logP of the compound i.e. positive logP compounds (indicating more lipophilic compounds) might be expected to have higher IC50s (& greater ratios) on the IonWorks Barracuda auto system since such closed well systems have been shown to give higher IC50 for lipophilic compounds (like terfenadine). This raised IC50 is due to the lipid membranes of cells that aren’t washed away from the recording site in a closed well system acting as ‘sinks’ for lipophilic compounds thus reducing the free concentration of compound (Bridgland-Taylor et al., JPTM, 2006).

I suggest the following areas might be worth further thought & consideration by the authors, potentially addressed to the editors & reviewers in their responses, but leave it to their discretion to directly address the reader in the MS text:

A. Line 113: “To support data transparency, the original electrophysiology records, detailed cell culture procedure, and supplemental materials for the present study are available for download at: https://osf.io/g3msb/.” – A v nice feature of PLoS ONE, however I have tried accessing this but various login attempts have failed thus far, it may just be my connection &/or login details (e.g. I’m yet to receive a confirmation email for the login).

B. Line 163: Internal & external recording solutions are equimolar at ~290 mOsM. Routinely, many labs use solutions that have internal 10-20 mOsM lower than external, any reason why equimolar is used?

C. Line 167: “When Ba2+ was used as the charge carrier, the external solution contained (in mM): 137 NaCl, 4 KCl, 4 BaCl2, 1 MgCl2, 10 HEPES, 10 glucose; pH adjusted to 7.4 with 5M NaOH; ~290 mOsM.” Ba2+ is higher than Ca2+ (1.8 mM) coupled with Ba2+ being conducted at higher rates than Ca2+ will lead to bigger Ba2+ currents than Ca2+, meaning bigger V-offset errors (& see E. below). There’s also the possibility that higher levels of charge carrying ion may have effects on the pore &/or selectivity filter, which could potentially confound the Ba2+ vs Ca2+ effects the authors were also investigating.

D. Line 186: “Signals were filtered at 3 or 10 kHz and sampled at 10 kHz.” Based on the Nyquist frequency sampling frequency rate, f, must be filtered at < f/2 to avoid signal aliasing i.e. sampling the signal at 10 kHz would need to be filtered at < 5 kHz, otherwise you risk distorting the recorded current & kinetics. In other words, your recordings filtered at 3 kHz are fine, but you have introduced signal aliasing in recordings filtered at 10 kHz.

E. Lines 365-422: The currents recorded are v big, starting at several nA at the start of recordings & are still ~ -2.2nA after 91-100 traces. The authors do a good job of defining & explaining the effects of recordings with & without 80% Rs compensation. However, with those large currents & large V-offset errors shown by the 80% Rs compensation analyses it suggests that for future work reducing these V-offsets further still would be worthwhile by:

a. Increasing the Rs-comp >80%

b. Reducing current sizes by either substituting non-conducting ions (though this then introduces non-physiological ions to your recordings) or titrating channel expression by changing tetracycline induction protocols (e.g. tet conc., post-tet culturing conditions & timings) & this may even be partially understood based on the three tetracycline induction protocols used (see 1. above)

F. Line 500: “For norbuprenorphine, there was no difference in the IC50s for ICa-step and ICa-ramp, suggesting a preference for open channel block when Ca2+ was used as the charge carrier.” I may have misunderstood, but feel that the lack of difference between step & ramp pharmacology could also suggest a closed state blocking effect.

Reviewer #3: Ren et al., describe how different experimental factors such as temperature, charge carrier and channel activation protocols affect pharmacology of the CaV1.2 calcium channel. Their stated main goal was to determine temperature dependence of channel block by different drugs.

The authors use a “step-step-ramp” voltage protocol to elucidate for a potential state-dependent blocking effect of calcium channel antagonists, various selective and non-selective opioid receptor agonists, and a muscarine receptor antagonist.

Drugs were tested under near physiological temperature vs. room temperature conditions and under different charge carrier conditions (Ca2+/Ba2+) in CHO cells stably expressing β2a and α2δ1. The human calcium channel hCaV1.2 was under the influence of a Tetracycline dependent promotor system.

The experimental key findings were a current rundown phenomenon under barium and calcium conditions, exquisite temperature sensitivity and a requirement for a high degree of series resistance compensation to optimize voltage command.

The group describes a temperature dependent blocking (Verapamil) and also facilitating effect (Methadone). Drugs seemed to show an activity dependent effect which was revealed by voltage ramps. An enhanced drug dependent preference was also observed by using barium over calcium. Overall, the blocking effect and slope from determined concentration-responses were shallow, temperature dependent and differed upon application of step vs. ramp.

The authors also conclude that the observed drug effects on CaV1.2 channels appear more complex than simple pore block mechanism.

Major General Concern:

In general, the authors chose to investigate a variety of very complex biophysical properties (temperature dependent, electrophysiological and ion dependent (Ba/Ca) properties) that affect CaV1.2 channel pharmacology. It would be more advisable to independently describe these different biophysical effects on CaV1.2 independent and hence in more detail and not just as a simple observation since electrophysiological properties of CaV1.2 are known to be complex and complicated, particularly when drug testing and biophysical aspects are involved.

Major Specific Concerns:

1. The step/ramp protocol is very complex yet it is not physiological and doesn’t allow to easily dissect and define effects of drugs on various parameters such as steady state inactivation or degree of pore block versus being gating modifiers (Striessnig et al., 2015).

2. A difference in pharmacological outcomes using calcium channel blockers have been described extensively by many groups within the last decades and frequency and use dependent binding characteristics of CCBs are nothing new per se. Also, the effect on binding through different charge carriers (Ba/Ca) was described in the past (e.g., Hering et al. 1997, Beyl et al., 2007, Shabbir et al., 2011, Sanguinetti & Kass 1984, Sokolov et al., 2001, Liao et al., 2007, to name just a few).

3. The Cav1.2 channel run down phenomenon is under investigation since 30 years and not understood at all. What is known, the effect is clearly charge carrier (Ba/Ca) and frequency dependent. Furthermore, calmodulin seems to be involved in this process. The here used experimental approach is too simplistic and superficial to properly describe the rundown phenomenon. There is clear evidence that Barium used a charge carrier does not induce CaV1.2 channel rundown.

4. Ren et al. author state that experiments required a high degree of series resistance compensation to optimize voltage control. However, they do not give any numerical examples of the series resistance. How large is the series resistance (Rs) and how large is the measured membrane resistance (Rm) in their experiments? How do these parameters look like at the beginning and at the end of the recordings?

These physical entities are important for an estimation of the approximate membrane potential (Vm) of the cell and the used command potential (Vcom). Furthermore, the authors state that the series resistance is a factor that impacts pharmacology. How so? Please describe this in more detail. What do you mean by that?

5. The maximal peak current cannot be the same (0mV) for barium and calcium due to the different conductivity. An IV plot would proof that. Please explain.

6. Verapamil blocks state dependent and based on the holding potential. This is well known (Nawrath 1997, Freeze et al., 2006).

7. 10 mM natrexon is so far from any relevant drug concentration that related findings are not interesting

8. To strengthen the work and findings, one exemplary time-course of all used drugs plus the additional control (Verapamil/Diltiazem) at the end of the experiment should be shown in the manuscript (particularly for Methadone, Tolterodine, Naltrexone, Naloxone, Norbuprenorphine and Buprenorphine).

6. PLOS authors have the option to publish the peer review history of their article (what does this mean?). If published, this will include your full peer review and any attached files.

Reviewer #1: No

Reviewer #2: **Yes: **Damian C. Bell

Reviewer #3: No

---

## [Author Response · Author response to Decision Letter 0]

14 Jul 2022

Please see submitted word doc "Response to Reviewers" as there are figures in that document.

Reviewer #1: The study from Ren and colleagues performs a dissection of key experimental factors affecting the outcome of pharmacological studies on Cav1.2 currents in an expression system. Up to 8 drugs are studied.

----- Lack of novelty -----

The study is well written, detailed in its methodology, very transparent and it is very difficult to argue with in its conclusions. This said, in my personal opinion the main weakness of the study is that it lacks scientific novelty: it was already known that temperature, degree of Rs compensation, charge carrier and the voltage-clamp protocol used are factors that affect the outcome of experiments.

The goal of this study was not to identify novel mechanisms that impact CaV1.2 channel pharmacology. Rather, we wanted to understand the extent to which each of the well-known experimental factor impacts drug block potencies for CaV1.2 channels. As we stated in the manuscript, we were motivated to understand why Crumb et al., 2016 and Li et al., 2018 reported large IC50 differences for the same clinical drugs (up to 1743X difference as summarized in Table 1) in spite of using similar voltage waveforms to mimic a ventricular action potential and presented at the same frequency. From the regulatory perspective, such magnitude of drug potency variability between laboratories makes it challenging to apply nonclinical ion channel data to predict or even explain clinical outcomes, thereby hindering the use of nonclinical ion channel data for decision-making. Therefore, we saw value in establishing the maximum variation in IC50 resulting from using Ba2+ vs. Ca2+, recording at RT vs. near PT, and analyzing drug effects at the step vs. the ramp current, as was done by the Crumb and Li papers. During the course of the study, we learned of additional factors that impacted our data but were not anticipated a priori based on literature research. Specifically, we did not anticipate that a couple of degrees of temperature oscillations due to the design and negative feedback nature of the temperature controller to cause dramatic fluctuations in the amplitudes of the Ca2+ and Ba2+ currents (Fig. 2). We knew that compensating for Rs was important to optimize voltage control in patch clamp experiments, but did not anticipate that it would have such a dramatic effect on the step current since based on the literature, we did not anticipate that current activation at the physiological temperature to be as fast (Fig. 3). Finally, while rundown of CaV1.2/L-type Ca2+ current has been mentioned in the methods section for some published studies, this phenomenon was not well-documented, and we were surprised by the magnitude and extent of rundown in our recordings even when Ba2+ was used (Fig. 1).

Insights gained from this study has shaped our expectations of CaV1.2 data and conduct of this assay. Results from this study have supported patch clamp protocol recommendation by the FDA Cardiac Safety Studies Interdisciplinary Review Team (CSS-IRT) for conducting CaV1.2 studies for proarrhythmia risk assessment (including the charge carrier to use and how drug effects are to be quantified; described below). We hope that this study will inform safety pharmacologists of factors to consider when setting up experimental protocols and analyzing patch clamp data from the CaV1.2 assay (to account for temperature oscillation during the recording, ensure adequate voltage control, and consider whether a Hill coefficient of 1 could be applied as a data quality criterion). We consider our study results as novel and important for the field of safety pharmacology, and humbly disagree with Reviewer 1 that this study lacked novelty. In spite of this disagreement, we sincerely thank Reviewer 1 for taking the time to comment on our manuscript and to pinpoint key areas that need improvement. 

----- Recommendations at the end of the paper are missing -----

My most constructive suggestion to improve the manuscript’s value is as follows: In my opinion, the manuscript’s call to standardize conditions among laboratories is fully justified. However, this corollary is not accompanied by EXPLICIT unifying criteria or explicit protocolary recommendations proposed by the authors that could help other laboratories to standardize such conditions. In the absence of such explicit recommendations, just pointing to the problem (which was known, see "Lack of novelty" above) is not a solution… In summary: how does this study contribute to solving the problem? 

To support proarrhythmia risk assessment, the FDA CSS-IRT has posted a document regarding the recommended voltage protocols for cardiac ion channels, including CaV1.2 channels, on its website (https://www.fda.gov/media/151418/download). The voltage protocol, stimulation frequency, compositions of the internal and external solutions (the latter using Ca2+ as the charge carrier), and data analysis method are the same as those used in this manuscript. 

For cardiac safety assessment, drug developers and regulators are following the guidelines released by the International Council for Harmonisation for Technical Requirements for Pharmaceuticals for Human Use: ICH S7B for nonclinical and ICH E14 for clinical studies. The protocol used in this manuscript is consistent with the best practice recommendations for patch clamp characterization of test article’s impact on cardiac ion channels (see newly released ICH S7B Q&A 2.1 guideline; https://database.ich.org/sites/default/files/E14-S7B_QAs_Step4_2022_0221.pdf; p. 31 – 35). 

We have revised the manuscript to include the above information (lines 675 – 686).

While some of such recommendations are obvious (e.g. compensate Rs to 70-80%), others are not (e.g. how-to best control temperature to reduce variability? statistics?). 

Regarding the impact of data variability due to temperature fluctuations for near PT recordings, one option is to further optimize temperature control, and another is to account for data variability / experimental uncertainty when interpreting the likelihood of direct CaV1.2 channel interactions by the test article. Regarding uncertainty estimation, please refer to ICH S7B Q&A 2.1 training material for details (https://database.ich.org/sites/default/files/ICH_E14-S7B_TrainingMaterial_2022_0407.pdf; p. 42). 

Topics that come to my mind and that could be addressed explicitly include:

(1) All data considered… is it more advisable to perform a step protocol or a step-step-ramp protocol while testing a given compound, from the points of view of variability and accuracy? In other words, which protocol has the highest changes of being reproduced among labs with the lowest variability, while still obtaining the proper drug potency and IC50?

We do not have sufficient information to comment on which voltage protocol yields the most reproducible data. Inter-laboratory data variability for CaV1.2 pharmacology attributed to voltage protocol alone without other differences in experimental or data analysis procedures is unclear based on literature search and is a topic currently under investigation. Regarding “proper drug potency and IC50”, we interpret this to mean improved translation of nonclinical ion channel data to clinical outcomes. Again there is insufficient empirical evidence at the moment to answer this question directly. Nonetheless, the voltage protocol used in this manuscript was designed to mimic a ventricular action potential paced at an adequate rate to assuage the concern of missing block that is frequency dependent. Therefore, it is reasonable to think that IC50 values obtained using this protocol would yield better translation to clinical outcomes than those obtained using a less “physiologically relevant” protocol. 

(2) Other protocols that should be considered? For example, in the study limitations the AP-clamp protocol is considered. What about 2-pulse protocols for recovery from refractoriness and/or inactivation properties?

Insights regarding drug-channel interaction characteristics are certainly worthwhile and may facilitate development of mechanistic in silico ventricular myocyte models to predict drug effects on cardiac action potentials. Consideration of additional protocols beyond those used to estimate IC50 values is a possibility recognized in the “Context of Use” section of the voltage protocol document provided by the FDA CSS-IRT. 

(3) Any proposal to better control the temperature such as a given chamber of definite dimensions, flow rate, etc? (the apparatus that works best is already mentioned). Any material that works best to avoid drug binding to it? 

We have described two types of temperature controlling devices / methods. For smaller bath volumes like ours (~0.5 mL), changes in the flow rate can produce a couple of degrees Celsius of temperature fluctuations. Our experience is that heating the chamber bottom for our gravity-fed perfusion system provides more stable temperature. We have no firsthand experience using setups with larger bath volumes to comment on temperature stability. To maintain flow rate, an option is to use a perfusion pump. However, in our brief use we noted perfusion noise in the recordings and did not further optimize perfusion pump system for our patch clamp studies. 

Ongoing research in our laboratory shows that nonspecific binding is drug-specific, as are materials that drugs bind to. Concentration verification using solution samples collected from the recording chamber during patch clamp recordings offers the best approximation of the drug concentrations exposed to the recorded cells (lines 602 – 618).

(4) When working at room temperature, can suitable correction factors be found or, because of additional issues such as differential drug binding at different temperatures, there is no point? How does this vary among drugs, given their known mechanism(s) of action? 

This study assessed methadone and verapamil at RT and near PT. No correction factor was found that can be applied to RT data to approximate near PT results. Temperature impact is drug-specific – a conclusion that is consistent with the literature data for nitrendipine, diltiazem, nifedipine, and flavoxate (see lines 482 – 504). Reviewer 1 asked a very important question regarding drug-channel interaction characteristics in relation to temperature-dependent pharmacology. As far as we know no generalization may be formed at the moment.

(5) When not addressing mechanistic studies (e.g. VDI vs. CDI in the presence of a drug), should results in the presence of calcium (as charge carrier) be preferred to those in barium because they reflect more faithfully the cellular physiology? 

This is the current thinking.

(6) How should data in expression systems be compared vs. data in, for example, ventricular myocytes or human iPSC-derived cardiomyocytes? It is important to acknowledge that myocytes experience a quantitatively strong calcium-dependent inactivation from SR calcium being released, as compared to expression systems where such SR release does not exist.

The current thinking is that data from the expression system offers a quick assessment of direct drug interactions with cardiac ion channel subunits expressed in the cell line, while the use of myocytes as a model system allows for assessing mechanisms that cannot be revealed by the expression system as Reviewer 1 pointed out. This idea is reflected in the newly released ICH S7B Q&A 2.2 – 2.5 (https://database.ich.org/sites/default/files/E14-S7B_QAs_Step4_2022_0221.pdf; p. 35 – 38). 

(7) Should laboratories provide WB data on the expression of subunits vs. a well-characterized, commercial cell line, available to all labs?

We interpret this question to mean laboratory-generated vs. commercially available cell line. There is currently insufficient information to favor one choice over the other. We would caution against recommending standardizing experimental factors before understanding each’s individual impact on pharmacology, for concerns of adding burden to otherwise difficult experiments while producing minimal impact on data. 

I have to add 2 extra recommendations, which are not very clearly stated in the methods section:

(8) Any recommendation on confidence intervals when performing a fit?

In this manuscript we used 95% CI. The current recommendation is to clearly define uncertainty parameter and to provide source data to allow for independent analysis.

(9) Any recommendation about finding outliers, given the strong inter-cell variability? For example: should outliers be considered prior to drug administration, after computing an inhibition ratio, or both?

None at the moment. We did not observe clear outliers. 

----- Readability -----

Perhaps for its amount of detail (which favors reproducibility), the paper does sometimes flow very slowly, and this could discourage readers. In my opinion, it may be best if authors transferred some of their written results to tables in order to summarize (so that readers remain focused).

We appreciate this feedback. To avoid redundancy we have already combined the results and discussion sections together. Upon rereading this manuscript we were unable to find sections that would be best summarized as tables. 

----- Comparison with previous literature is performed qualitatively -----

I missed a table comparing the estimated IC50 values in this study with those values from prior literature. The 2 studies cited (refs. 5 and 6) are extreme examples for their differences, but there should be more studies to compare with. Where (within the known literature range) do the current IC50s dwell?

In fact, I missed a table or paragraph describing what was known about each of the tested drugs before this study, as a part of the introduction. Interested readers who are not experienced may get easily lost or fail to understand if the drug mechanisms proposed in this study have been already described or not.

We made Table 2 to summarize literature data. In reading the manuscript we were unable to fit this table into the introduction section as Reviewer 1 envisioned. Instead we refer to it in the “Results and Discussion” section. Please see lines 592 – 598 in the manuscript text to refer to these results.

----- Question about temperature dependency -----

In the traces shown in Figure 2B, the curve in ICa-ramp amplitude appears to be time-shifted (i.e. delayed) vs. the temperature curve, rather than being totally unrelated to temperature. Have the authors considered the possibility of such delay and whether it applies to cells where no current-temperature relationship was observed? A correlation with some temporal shift between the curves could be a good way to demonstrate (presence or absence) this behaviour

We would like to clarify that the relationship between ICa-ramp with temperature is inconsistent across cells (see Supplemental Fig. 3A-E; lines 347 – 349), and there was no dominant phenotype based on the 6 cells tested. For example, the plot below shows the cell in Supplemental Fig. 3C with linear ICa-ramp vs. temperature relationship. Given the mixed phenotype observed, we respectfully request for Reviewer 1’s understanding of our preference to not further analyze this complex phenomenon. Temperature-dependence of CaV1.2 channel gating warrants further dedicated investigation. 

----- Info missing in methods -----

Please state lag (microseconds) when applying Rs compensation.

The MultiClamp 700B the Rs compensation bandwidth control replaces the “lag” control on earlier Axon amplifier series: Bandwidth = 1 / (2 * π * Lag). We used the default Rs correction bandwidth of 1.02 kHz, which is equivalent to a lag value of 156 μs. This information is now in the manuscript (lines 195 – 198).

Please state R(input) and Rs tolerance limits (i.e. for R(input) and Rs to be considered constant during an experiment: for example “R(input) and Rs were considered constant if they did not change by more than XXXXX percent during the experiments”.

Please see topics (8) and (9) above and provide additional information about the fitting procedure and the identification of outliers

Rs was only measured once, roughly 2 min following whole cell dialysis after the concern of rapid resealing had passed. The range of starting Rs for this study is 4.5 ± 0.1 MΩ (n = 295). This information is now added to the methods, and a spreadsheet with individual cells’ Rs values, organized per recording temperature/charge carrier used, will be released through https://osf.io/g3msb/ upon manuscript acceptance. Whether Rs changed dramatically or not throughout the experiments was based on offline analysis – assessment of current kinetics (elicited by the 0 mV step) and voltage at which ramp peak occurred across time (e.g., see Fig. 3C, 3G, Supplemental Fig. 1G), stability of the holding current at -80 mV, and stability of Rinput (e.g., see time course plots in Fig. 2C, 2H, 3C, 3G, Supplemental Fig. 1C, 1E, 2C). For Rinput, stability was based on visual inspection of the time course plots for every cell. At this point we prefer to avoid giving specific tolerance limits for changes in these parameters, since the impact on pharmacology will depend on other factors such as the % of Rs compensation, starting Rs value, and Rinput value. 

----- Minor -----

By reading the title, I could not understand the words “current region of analysis”. It took a while, until the authors described the “step-step-ramp” protocol, that the title made full sense to me. I would suggest rewording of that part.

We have changed our title to the following: 

Experimental factors that impact CaV1.2 channel pharmacology – effects of recording temperature, charge carrier, and quantification of drug effects on the step and ramp currents elicited by the “step-step-ramp” voltage protocol

Reviewer #2: Temperature dependence of ion channel pharmacology, particularly at physiological temperatures, is very important. However, despite this importance it is under investigated, so I applaud the authors for their thorough examination in Cav1.2 ion channels. The challenges, some of the solutions found & highlighting other areas that require further work will be useful to the field, & will help to get greater consistency & physiologically relevant comparisons between labs in critical areas like cardiac ion channel safety pharmacology.

We thank Reviewer 2 for these encouraging words, and for providing comments to improve our manuscript.

I suggest the following Qs & areas should be addressed within the MS to help improve & explain the findings for the reader:

1. Line 29: “Three protocols were used for tetracycline induction.” Why three & was any one protocol preferred over others e.g. pros/cons that may explain the choice?

Three protocols were used to accommodate staff schedule for this research. Having cells available at all times for recording was especially important during COVID-19, as each staff was granted limited hours per day / limited days per week to come to the campus to conduct research. 

We are currently using the 3rd protocol for CaV1.2 experiments. After induction of the α subunit our cells adopt a very flat morphology, making patching and maintaining long lasting recordings more challenging. Cells detached on the day of the experiment (protocol 3) are easier to patch and more stable due to the more rounded morphology. Note that the use of different cell culture procedures did not impact pharmacology in this study.

We have provided this information in the revised manuscript (lines 139 – 142).

2. Line 48: “Two [temperature] controller models were used …” Nice explanation of the two controllers. But seems that in both intra-bath temperature variations & in overall temperature consistency that the TC2BIP was superior. Yet it seems despite this that the TC-344C was used (line 318: “Therefore, temperature sensitivity of the CaV1.2 current was examined. These experiments were conducted using setups with the TC-344C controller …”). Why?

Our impression was that CaV1.2 current amplitude fluctuations were more common when recordings were performed with the rigs that had TC-344C temperature controller than with TC2BIP. Hence we opted to characterized how bath temperature fluctuations may be related to CaV1.2 amplitude fluctuations using TC-344C. 

Please see line 334 for this updated information.

3. Line 272: “The normalized ICa-step at the 150th trace was 0.42 ± 0.04 relative to the 1st trace (Fig. 1C); the normalized ICa-ramp, 0.44 ± 0.06 (Fig. 1D). The normalized IBa-step was 0.34 ± 0.04 (Fig. 1E); the normalized IBa-ramp, 0.32 ± 0.04 (Fig. 1F).” Are the bigger post-run-down currents for Ca2+ (~0.43 nA) vs Ba2+ (~0.33nA) significantly different? If so, does that suggest a Ca-dependent reduction in run-down? (It seems a little counter-intuitive, since Ba obviously reduces current Ca-dependent inactivation, as explained by the authors. However, such inactivation is temporary & mostly recovered between depolarising pulses, whilst run-down is pseudo-permanent & cumulative over the time-course of such experiments.)

We performed a basic statistical analysis of the differences in the amplitude of the 150th trace relative to the 1st trace. A t-test revealed no significant difference for comparisons between ICa-step and IBa-step and between ICa-ramp and IBa-ramp (p > 0.05). Please see lines 272 – 273 and 286 – 288 for this updated information.

4. Lines 595-606: V good description & explanation of possible solutions & analysis for run-down currents. Another possible issue that could be highlighted here is that the effects recorded after run-down might be vs a specific channel population, such as specific subunit composition(s) &/or phosphorylation state(s) mentioned in the third limitation (lines 607-613) i.e. waiting to get a plateau current after run-down may mean you’ve skewed your recordings to a subpopulation of channels &/or phosphorylation state.

We thank Reviewer 2 for raising this possibility. Within a given cell, the ramp-to-step ratio remained stable as rundown occurred (Fig. 1G and 1H). Additionally, within the cell the I-V relation, approximated based on the voltage at which the ramp current peaked, does not change throughout the recording (Fig. 3C; Supplemental Fig. 1G). The lack of evidence of changes in voltage-dependence of channel gating argues against skewing of recording to a subpopulation of channels by waiting for quasi-steady state to test pharmacology. 

5. Line 643: “… multiple binding poses (i.e., diltiazem), multiple binding sites (i.e., nitrendipine, roscovitine), or through even allosteric mechanisms (i.e., diltiazem; naltrexone, Fig. 7).” Even needs deleting.

Done.

6. Line 659: Table 1 – it might be worth adding an extra column to this table to give logP (measure of lipophilicity) of these compounds. It would be interesting to see if these ratios correlate to the logP of the compound i.e. positive logP compounds (indicating more lipophilic compounds) might be expected to have higher IC50s (& greater ratios) on the IonWorks Barracuda auto system since such closed well systems have been shown to give higher IC50 for lipophilic compounds (like terfenadine). This raised IC50 is due to the lipid membranes of cells that aren’t washed away from the recording site in a closed well system acting as ‘sinks’ for lipophilic compounds thus reducing the free concentration of compound (Bridgland-Taylor et al., JPTM, 2006).

We added a column reporting logP values to Table 1 (also see lines 615 – 722). There is no relationship between the ratio (max-to-min IC50s reported by the two studies) and logP values for these drugs. 

I suggest the following areas might be worth further thought & consideration by the authors, potentially addressed to the editors & reviewers in their responses, but leave it to their discretion to directly address the reader in the MS text:

A. Line 113: “To support data transparency, the original electrophysiology records, detailed cell culture procedure, and supplemental materials for the present study are available for download at: https://osf.io/g3msb/.” – A v nice feature of PLoS ONE, however I have tried accessing this but various login attempts have failed thus far, it may just be my connection &/or login details (e.g. I’m yet to receive a confirmation email for the login).

We have procured the URL link to host the CaV1.2 channel dataset and other pertinent information shared in this manuscript. However, the contents are only publicly available following manuscript acceptance for publication. 

B. Line 163: Internal & external recording solutions are equimolar at ~290 mOsM. Routinely, many labs use solutions that have internal 10-20 mOsM lower than external, any reason why equimolar is used?

No reason. We tested these internal / external solution combinations and found them to work at the onset of the study, hence continued with them.

C. Line 167: “When Ba2+ was used as the charge carrier, the external solution contained (in mM): 137 NaCl, 4 KCl, 4 BaCl2, 1 MgCl2, 10 HEPES, 10 glucose; pH adjusted to 7.4 with 5M NaOH; ~290 mOsM.” Ba2+ is higher than Ca2+ (1.8 mM) coupled with Ba2+ being conducted at higher rates than Ca2+ will lead to bigger Ba2+ currents than Ca2+, meaning bigger V-offset errors (& see E. below). There’s also the possibility that higher levels of charge carrying ion may have effects on the pore &/or selectivity filter, which could potentially confound the Ba2+ vs Ca2+ effects the authors were also investigating.

If Rs for different recordings were the same, then we agree that larger current would be associated with larger voltage error, and recommend compensating at the highest possible degree that is feasible and practical for long lasting pharmacology experiments. 

Reviewer 2’s 2nd comment is a possibility. Given the spread of data in Fig. 5 and the magnitude of IC50 differences shown in Fig. 6 between Ca2+ and Ba2+ experiments, we speculate that the impact on pharmacology due to using 4.0 mM vs. 1.8 mM would be small and difficult to resolve given the cell-to-cell data variability due to other experimental factors addressed in this manuscript. 

D. Line 186: “Signals were filtered at 3 or 10 kHz and sampled at 10 kHz.” Based on the Nyquist frequency sampling frequency rate, f, must be filtered at < f/2 to avoid signal aliasing i.e. sampling the signal at 10 kHz would need to be filtered at < 5 kHz, otherwise you risk distorting the recorded current & kinetics. In other words, your recordings filtered at 3 kHz are fine, but you have introduced signal aliasing in recordings filtered at 10 kHz.

We agree with Reviewer 2’s comment, and would like to clarify that filtering at 10 kHz was a mistake by a research fellow for a brief period of time, and that this mistake had no impact on pharmacology. In the voltage protocol document shared by the FDA CSS-IRT, the recommendation is filtering at 3 kHz and digitizing at 10 kHz. 

E. Lines 365-422: The currents recorded are v big, starting at several nA at the start of recordings & are still ~ -2.2nA after 91-100 traces. The authors do a good job of defining & explaining the effects of recordings with & without 80% Rs compensation. However, with those large currents & large V-offset errors shown by the 80% Rs compensation analyses it suggests that for future work reducing these V-offsets further still would be worthwhile by:

a. Increasing the Rs-comp >80%

b. Reducing current sizes by either substituting non-conducting ions (though this then introduces non-physiological ions to your recordings) or titrating channel expression by changing tetracycline induction protocols (e.g. tet conc., post-tet culturing conditions & timings) & this may even be partially understood based on the three tetracycline induction protocols used (see 1. above)

We thank Reviewer 2 for this comment. The ramp current amplitude in external Ca2+ was on average only 30% that of the step current (Fig. 1G). Additionally, both the step and ramp currents ran down by about 60 – 70% at quasi-steady state (Fig. 1C – F). Because we compared drug effect on the step vs. ramp current, having a larger step current is a trade-off to obtain convincing pharmacology data on the ramp current. 

Compensating Rs at 80% and not higher was balanced by our need to hold the cell for a long period of time for rundown to slow down followed by drug applications. With even a higher degree of compensation, a small decrease in Rs would lead to oscillation, likely leading to losing the recording. We recommend compensating Rs at as high of a level as possible given the specific cell line / current characteristics and amplifier used by each laboratory. 

F. Line 500: “For norbuprenorphine, there was no difference in the IC50s for ICa-step and ICa-ramp, suggesting a preference for open channel block when Ca2+ was used as the charge carrier.” I may have misunderstood, but feel that the lack of difference between step & ramp pharmacology could also suggest a closed state blocking effect.

We agree with Reviewer 2 regarding this possibility. However, our understanding is that most drugs block CaV1.2 channels in open and/or inactivated state, not closed state. Accordingly, we interpret the lack of difference between the step and ramp current inhibition to indicate a preference for open state block. 

Reviewer #3: Ren et al., describe how different experimental factors such as temperature, charge carrier and channel activation protocols affect pharmacology of the CaV1.2 calcium channel. Their stated main goal was to determine temperature dependence of channel block by different drugs.

We wish to clarify that our goal was to study how each experimental factor mentioned in the study impacts pharmacology. Please also see our response to Reviewer 1 on p. 1 – 2 of this cover letter.

The authors use a “step-step-ramp” voltage protocol to elucidate for a potential state-dependent blocking effect of calcium channel antagonists, various selective and non-selective opioid receptor agonists, and a muscarine receptor antagonist.

Drugs were tested under near physiological temperature vs. room temperature conditions and under different charge carrier conditions (Ca2+/Ba2+) in CHO cells stably expressing β2a and α2δ1. The human calcium channel hCaV1.2 was under the influence of a Tetracycline dependent promotor system.

The experimental key findings were a current rundown phenomenon under barium and calcium conditions, exquisite temperature sensitivity and a requirement for a high degree of series resistance compensation to optimize voltage command.

Indeed, these are characteristics of CaV1.2 current under our experimental condition. 

The group describes a temperature dependent blocking (Verapamil) and also facilitating effect (Methadone). Drugs seemed to show an activity dependent effect which was revealed by voltage ramps. An enhanced drug dependent preference was also observed by using barium over calcium. Overall, the blocking effect and slope from determined concentration-responses were shallow, temperature dependent and differed upon application of step vs. ramp.

The authors also conclude that the observed drug effects on CaV1.2 channels appear more complex than simple pore block mechanism.

We thank Reviewer 3 for this summary. These findings are important to support standardization of experimental protocols for generating CaV1.2 channel pharmacology intended for proarrhythmia risk assessment.

Major General Concern:

In general, the authors chose to investigate a variety of very complex biophysical properties (temperature dependent, electrophysiological and ion dependent (Ba/Ca) properties) that affect CaV1.2 channel pharmacology. It would be more advisable to independently describe these different biophysical effects on CaV1.2 independent and hence in more detail and not just as a simple observation since electrophysiological properties of CaV1.2 are known to be complex and complicated, particularly when drug testing and biophysical aspects are involved.

This research was motivated to understand factors that might have contributed the large variability in IC50s reported by Crumb et al., 2016 and Li et al., 2018 for the same drugs. Therefore, we have resorted to using a similar voltage protocol. We wish to point out that this protocol has appropriate elements of a ventricular action potential, hence is consistent with the best practice recommendations per ICH S7B Q&A 2.1 guideline mentioned above for experiments to support cardiac safety assessment (see p. 2 of this cover letter for our response to Reviewer 1’s 2nd comment). We agree that additional experiments are needed to gain further mechanistic insights regarding of how charge carrier, temperature, and channel state impact drug binding to CaV1.2 channels. Regrettably, these experiments are beyond the scope of the current study. 

Major Specific Concerns:

1. The step/ramp protocol is very complex yet it is not physiological and doesn’t allow to easily dissect and define effects of drugs on various parameters such as steady state inactivation or degree of pore block versus being gating modifiers (Striessnig et al., 2015).

We stated the reason of using this voltage protocol above. Regarding physiological relevance, for cardiac safety assessment, the newly released ICH S7B Q&A 2.1 guideline lists 7 best practice recommendations for patch clamp assays designed to evaluate drug potency on affecting cardiac ionic currents using overexpression cell lines. Point #2 states that the voltage protocol should approximate the appropriate elements of a ventricular action potential (p. 31 of the Q&A document). In our opinion, the current protocol satisfies this point in that the voltage protocol included rapid depolarization, appropriately long duration of depolarization above 0 mV, and terminal voltage ramp to mimic repolarization in myocytes.

2. A difference in pharmacological outcomes using calcium channel blockers have been described extensively by many groups within the last decades and frequency and use dependent binding characteristics of CCBs are nothing new per se. Also, the effect on binding through different charge carriers (Ba/Ca) was described in the past (e.g., Hering et al. 1997, Beyl et al., 2007, Shabbir et al., 2011, Sanguinetti & Kass 1984, Sokolov et al., 2001, Liao et al., 2007, to name just a few).

We agree with Reviewer 3’s comment. Crumb et al., 2016 and Li et al., 2018 used similar voltage protocols and the same stimulation frequency, presumably normalizing frequency- and use-dependent drug binding. Yet the resultant IC50s were still wildly different for the same drugs (see Table 1 of the manuscript). Prior studies assessed the impact of charge carriers also did not report such magnitude of pharmacology difference. The mismatch between the small impact on drug binding due to manipulating different experimental factors vs. the large magnitude of IC50 differences presented by the Crumb and Li papers thus prompted us to conduct the present study. 

3. The Cav1.2 channel run down phenomenon is under investigation since 30 years and not understood at all. What is known, the effect is clearly charge carrier (Ba/Ca) and frequency dependent. Furthermore, calmodulin seems to be involved in this process. The here used experimental approach is too simplistic and superficial to properly describe the rundown phenomenon. There is clear evidence that Barium used a charge carrier does not induce CaV1.2 channel rundown.

In our hands CaV1.2 current rundown was seen in every cell, including when Ba2+ was used as a charge carrier (Fig. 1). The choice of charge carrier did not impact % current remaining at the “quasi-steady state” level (see our answer to Reviewer 2’s question #3 on p. 7 of this cover letter). 

The rationale to characterize the time course of current rundown was that this could be a source of CaV1.2 pharmacology variability within our laboratory and across different laboratories, not to investigate the mechanisms that mediate current rundown. We agree that rundown is a complex phenomenon and we reviewed some of the literature on mechanism on p. 15 – 16 (lines 319 – 324).

4. Ren et al. author state that experiments required a high degree of series resistance compensation to optimize voltage control. However, they do not give any numerical examples of the series resistance. How large is the series resistance (Rs) and how large is the measured membrane resistance (Rm) in their experiments? How do these parameters look like at the beginning and at the end of the recordings?

These physical entities are important for an estimation of the approximate membrane potential (Vm) of the cell and the used command potential (Vcom). 

For experiments that compared the impact of Rs compensation on Ca2+ and Ba2+ current amplitudes, we did provide numerical values in the Fig. 3D and 3H and manuscript text (p. 19 – 20). This information was used to estimate voltage loss through Rs for the recorded cells reported on these pages. For transparency, the table below provides Rs (uncompensated total value) and Rinput values (assessed at -80 mV) values for all cells used in Fig. 3.

During the pharmacology experiments, we measure Rs once at the beginning of the experiment (after no more signs of rapid membrane resealing) and maintained 80% compensation throughout the recording to avoid gaps in the measurement of current amplitude. We did not measure Rs at the end of each recording. Whether Rs changed dramatically or not throughout the experiments was based on offline analysis – assessment of current kinetics (elicited by the 0 mV step) and voltage at which ramp peak occurred across time (e.g., see Fig. 3C, 3G, Supplemental Fig. 1G), stability of the holding current at -80 mV, and stability of Rinput (e.g., see time course plots in Fig. 2C, 2H, 3C, 3G, Supplemental Fig. 1C, 1E, 2C). 

Note that we have now included in the manuscript the Rs values for all pharmacology experiments as well (lines 198 – 200). This information as well as whole cell capacitance for individual cells will be released in a spreadsheet through https://osf.io/g3msb/.

Furthermore, the authors state that the series resistance is a factor that impacts pharmacology. How so? Please describe this in more detail. What do you mean by that?

The magnitude of voltage loss through Rs is related to the amplitude of the current elicited by the 0 mV step. During a pharmacology experiment, the size of the current is reduced with increasing drug concentration. Therefore, the magnitude of voltage loss is also reduced with increasing drug concentration (assuming constant Rs throughout the experiment). This means that current amplitude throughout the pharmacology experiment is not elicited by the same membrane voltage. 

We have attempted to minimize the changing voltage loss by minimizing Rs with 80% compensation.

5. The maximal peak current cannot be the same (0mV) for barium and calcium due to the different conductivity. An IV plot would proof that. Please explain.

We are unclear what Reviewer 3 was referring to in this statement. Our observation indeed was that Ba2+ current was larger than Ca2+ current, collectively speaking. For quantitative measures, please see lines 383, 385, 407, and 408. For cells used to measure kinetics of Ca2+ and Ba2+ currents, Ba2+ current elicited by the 0 mV step was nearly twice as large as Ca2+ current. 

Note that we have used 3 different tetracycline induction protocol for this manuscript, and we clearly stated that the size of the current is a function of tetracycline concentration used and induction time (lines 142 – 144). 

6. Verapamil blocks state dependent and based on the holding potential. This is well known (Nawrath 1997, Freeze et al., 2006).

We agree, hence our interpretation that the different IC50s obtained at the 0 mV step and the voltage ramp could reflect state-dependent block. 

7. 10 mM natrexon is so far from any relevant drug concentration that related findings are not interesting

Naltrexone on Ca2+ current was assessed in a prior study to understand the possibility of multi-cardiac ion channel block by this drug (Tran et al., 2020; https://journals.plos.org/plosone/article?id=10.1371/journal.pone.0241362), and in the process of obtaining these data we compared its effect on Ba2+ current mediated by CaV1.2 channels. This drug is interesting to us in the sense that larger ramp current could arise from a shift in the voltage that ramp peak occurred (i.e., by changing channel gating), as opposed to drug dissociating from the channel at +30 mV upon initial block at 0 mV. 

8. To strengthen the work and findings, one exemplary time-course of all used drugs plus the additional control (Verapamil/Diltiazem) at the end of the experiment should be shown in the manuscript (particularly for Methadone, Tolterodine, Naltrexone, Naloxone, Norbuprenorphine and Buprenorphine).

Done. Please see the new Supplemental Fig. 6 and 7 for Ca2+ and Ba2+ currents, respectively (also lines 524 – 525).

---

## [Decision Letter · Decision Letter 1]

1 Aug 2022

PONE-D-22-10448R1Experimental factors that impact CaV1.2 channel pharmacology – effects of recording temperature, charge carrier, and quantification of drug effects on the step and ramp currents elicited by the “step-step-ramp” voltage protocolPLOS ONE

Dear Dr. Wu,

Thank you for submitting your manuscript to PLOS ONE. After careful consideration, we feel that it has merit but does not fully meet PLOS ONE’s publication criteria as it currently stands. Therefore, we invite you to submit a revised version of the manuscript that addresses the points raised during the review process.

In particular the Authors should take note of the comments of Reviewers 1 and 3, in terms of additional discussion and recognition of experimental limitations.

We look forward to receiving your revised manuscript.

Kind regards,

Daniel M. Johnson, PhD

Academic Editor

PLOS ONE

Journal Requirements:

Reviewers' comments:

Reviewer's Responses to Questions

**Comments to the Author**

1. If the authors have adequately addressed your comments raised in a previous round of review and you feel that this manuscript is now acceptable for publication, you may indicate that here to bypass the “Comments to the Author” section, enter your conflict of interest statement in the “Confidential to Editor” section, and submit your "Accept" recommendation.

Reviewer #1: (No Response)

Reviewer #2: All comments have been addressed

Reviewer #3: (No Response)

2. Is the manuscript technically sound, and do the data support the conclusions?

Reviewer #1: Yes

Reviewer #2: (No Response)

Reviewer #3: No

3. Has the statistical analysis been performed appropriately and rigorously? 

Reviewer #1: Yes

Reviewer #2: (No Response)

Reviewer #3: I Don't Know

4. Have the authors made all data underlying the findings in their manuscript fully available?

Reviewer #1: Yes

Reviewer #2: (No Response)

Reviewer #3: Yes

5. Is the manuscript presented in an intelligible fashion and written in standard English?

Reviewer #1: Yes

Reviewer #2: (No Response)

Reviewer #3: Yes

6. Review Comments to the Author

Reviewer #1: --- Overall ---

The authors have provided a very detailed “response to reviewers”, which has clarified the concerns that I had about the study overall and helped strengthening a positive opinion on my part.

In going through the resubmitted version of the manuscript, the combination of results and discussion has greatly improved readability. Neophyte readers will greatly appreciate Table 2 as well.

--- Minor ---

Perhaps this is a personal point of view, but I feel that some of the detail provided in the “response to reviewers” has not been incorporated to the manuscript, and doing so could still add further value. Specifically,

(1) If the authors did not observe clear outliers in their data despite high cell-to-cell variability, please state so in the manuscript.

(2) If the authors found that literature information or empirical evidence are insufficient to address a certain question (e.g. specific recommendations about temperature control?) please state so in the manuscript.

Stating that information is insufficient is more helpful to readers, at least in my opinion, than stating nothing.

Reviewer #2: (No Response)

Reviewer #3: Although it is obvious that this work required extensive time commitment and the Authors clearly spent time and effort to address all concerns in their response and revision, the following serious concerns remain.

1. In most sample traces (e.g., Suppl Fig 6 and 7) the current run down is unusually strong (especially for Ba, for which most investigators find much less if at all run down over time). In a number of example traces, the run down is very variable over the time course. This variability and the very strong run down seems to us to preempt quantification of drug effects, which appear often smaller than the observed run down that is intrinsic to the authors’ measurements. One possibility is that this is due to a strong change of Rs over time. Because no time courses are presented for Rs, this possibility cannot be excluded.

2. Reversal potentials were not determined by testing over the whole voltage range but extrapolated from currents obtained for depolarizations between 0 and +20 mV (Suppl Fig 5). It is unclear why the reversal potentials were not directly measured. Also critically, for Ba should be closer to 50 mV and Ca closer to 80 mV than rather than 35 and 46, respectively.

7. PLOS authors have the option to publish the peer review history of their article (what does this mean?). If published, this will include your full peer review and any attached files.

Reviewer #1: No

Reviewer #2: **Yes: **Damian C. Bell

Reviewer #3: No

---

## [Author Response · Author response to Decision Letter 1]

29 Sep 2022

Please see cover letter. We have provided a graph to clarify our answer, and the graph could not be pasted here.

Reviewer #1: --- Overall ---

The authors have provided a very detailed “response to reviewers”, which has clarified the concerns that I had about the study overall and helped strengthening a positive opinion on my part.

In going through the resubmitted version of the manuscript, the combination of results and discussion has greatly improved readability. Neophyte readers will greatly appreciate Table 2 as well.

We thank Reviewer 1 for these encouraging remarks.

--- Minor ---

Perhaps this is a personal point of view, but I feel that some of the detail provided in the “response to reviewers” has not been incorporated to the manuscript, and doing so could still add further value. Specifically,

(1) If the authors did not observe clear outliers in their data despite high cell-to-cell variability, please state so in the manuscript. This statement is now included in the manuscript. Please see lines 656 - 657.

(2) If the authors found that literature information or empirical evidence are insufficient to address a certain question (e.g. specific recommendations about temperature control?) please state so in the manuscript. Stating that information is insufficient is more helpful to readers, at least in my opinion, than stating nothing. Please see lines 704 – 714. We have stated our experiences using the two temperature controllers, emphasizing that for gravity-fed perfusion systems (that could lead to changes in the flow rate due to changing solution levels in the reservoirs) and small/shallow bath chambers like ours, the model that heats the chamber bottom (on which the coverslip with cells directly sat) provided more stable bath temperature maintenance near PT. As our temperature fluctuations were related to changes in the flow rate, we have also speculated that the other temperature controller model may have worked well should a perfusion pump be used to deliver the recording solutions. Finally, we have provided two recommendations for experimenters interested in testing CaV1.2 pharmacology at near PT: 1) to understand the extent of temperature fluctuations per their rig designs, and 2) to capture temperature readout throughout the recordings (to enable offline analysis of temperature sensitivity should one be interested).

We have revised the manuscript by incorporating Reviewer 1’s two specific requests mentioned above. Reviewer 1 (and 2 and 3) had raised many important questions that we have addressed in the cover letter accompanying the first revision. However, as was pointed out then, the amount of information provided in this study already made the manuscript dense, which Reviewer 1 cautioned would discourage readers. We are concerned that by incorporating all the responses in the first cover letter into the second revision, we would further exacerbate the issue, especially given that addressing Reviewer 3’s two concerns in revision 2 also increased text and introduced new concepts. PLOS ONE publishes questions raised during the review process as well as authors’ point-to-point responses. Therefore, interested readers will have access to all the Q&As. We will also upload both cover letters to https://osf.io/g3msb/ associated with this manuscript. 

Reviewer #3: Although it is obvious that this work required extensive time commitment and the Authors clearly spent time and effort to address all concerns in their response and revision, the following serious concerns remain.

1. In most sample traces (e.g., Suppl Fig 6 and 7) the current run down is unusually strong (especially for Ba, for which most investigators find much less if at all run down over time). In a number of example traces, the run down is very variable over the time course. This variability and the very strong run down seems to us to preempt quantification of drug effects, which appear often smaller than the observed run down that is intrinsic to the authors’ measurements. One possibility is that this is due to a strong change of Rs over time. Because no time courses are presented for Rs, this possibility cannot be excluded.

We thank Reviewer 3 for this comment. The extent of Ca2+ and Ba2+ current rundown is cell-specific, and can be extensive as we have pointed out in the manuscript. We agree that this phenomenon is not anticipated a priori based on literature search, hence we tested 3 cell lines total with CaV1.2 channels that included β2 and α2δ1 subunits (see Sup. Fig. 2 for two additional cell lines). That IC50 could be underestimated due to rundown was stated in the original manuscript (lines 620 – 622). We believe that current rundown as well as its potential impact on estimating drug block potencies of CaV1.2 channels should be conveyed clearly and discussed, as this phenomenon should be considered when interpreting CaV1.2 data. 

We acknowledge that not having a time course plot of Rs throughout the recording for every cell is a limitation, and have included this point in the manuscript (see lines 629 – 640). In our previous study on NaV1.5 current, we measured Rs for every trace (Wu et al., 2019; https://pubmed.ncbi.nlm.nih.gov/31255744/). However, estimating and compensating Rs in between each trace took time, and doing so would have precluded our delivering voltage waveform at 5 s intervals (a recommendation of the ICH S7B Q&A 2.1 is to not miss frequency-dependent block, which would be a concern for pharmacology experiments if stimulating at an even slower frequency than the present study). Another option is to the pause recording during the middle of the experiment to obtain Rs measurement. However, interruption of pacing would then lead to a change in the current amplitude, thereby prolonging the recording time in the control solution. Ca2+ and Ba2+ currents recorded at near PT activate extremely fast, and a large change in Rs can be seen as a slowing of time-to-peak for the step current and a shift in the voltage at which ramp peak occurred (also see Fig. 3). Below are Ba2+ current traces (65, 75, 85, 95, and 105) from the buprenorphine cell shown in Supplemental Fig. 7, illustrating no change in the time-to-peak for the step current as well as the voltage at which the ramp peak occurred in spite of current rundown. 

We do not believe current rundown, including that seen in Ba2+ experiments, was due to strong and progressive change in Rs over time based on these offline analyses. Having said that, we thank Reviewer 3 for raising this logical concern. By making all electrophysiology records available through https://osf.io/g3msb/, we provide interested readers the opportunity to go over the original recordings to form independent conclusions, including mechanisms subserving rundown. 

2. Reversal potentials were not determined by testing over the whole voltage range but extrapolated from currents obtained for depolarizations between 0 and +20 mV (Suppl Fig 5). It is unclear why the reversal potentials were not directly measured. Also critically, for Ba should be closer to 50 mV and Ca closer to 80 mV than rather than 35 and 46, respectively.

We thank Reviewer 3 for this comment, and have now included our extrapolation method for the reversal potentials of the Ba2+ and Ca2+ currents as a limitation (see lines 640 – 645). In these experiments, the I-V relations were generated for a subset of cells to assess voltage control, by verifying that the currents increased in an incremental fashion and not “jump” to the maximum amplitude. Our focus was thus in the voltage range between -60 and -20 mV. We agree that for the purpose of assessing reversal potential, extending the voltage steps to beyond the reversal potential would offer direct measurement.

---

## [Decision Letter · Decision Letter 2]

19 Oct 2022

Experimental factors that impact CaV1.2 channel pharmacology – effects of recording temperature, charge carrier, and quantification of drug effects on the step and ramp currents elicited by the “step-step-ramp” voltage protocol

PONE-D-22-10448R2

Dear Dr. Wu,

We’re pleased to inform you that your manuscript has been judged scientifically suitable for publication and will be formally accepted for publication once it meets all outstanding technical requirements.

Kind regards,

Daniel M. Johnson, PhD

Academic Editor

PLOS ONE

Additional Editor Comments (optional):

Reviewers' comments:

Reviewer's Responses to Questions

**Comments to the Author**

1. If the authors have adequately addressed your comments raised in a previous round of review and you feel that this manuscript is now acceptable for publication, you may indicate that here to bypass the “Comments to the Author” section, enter your conflict of interest statement in the “Confidential to Editor” section, and submit your "Accept" recommendation.

Reviewer #1: All comments have been addressed

2. Is the manuscript technically sound, and do the data support the conclusions?

Reviewer #1: Yes

3. Has the statistical analysis been performed appropriately and rigorously? 

Reviewer #1: Yes

4. Have the authors made all data underlying the findings in their manuscript fully available?

Reviewer #1: Yes

5. Is the manuscript presented in an intelligible fashion and written in standard English?

Reviewer #1: Yes

6. Review Comments to the Author

Reviewer #1: (No Response)

7. PLOS authors have the option to publish the peer review history of their article (what does this mean?). If published, this will include your full peer review and any attached files.

Reviewer #1: No

---

## [Editor Report · Acceptance letter]

14 Nov 2022

PONE-D-22-10448R2 

Experimental factors that impact CaV1.2 channel pharmacology – effects of recording temperature, charge carrier, and quantification of drug effects on the step and ramp currents elicited by the “step-step-ramp” voltage protocol 

Dear Dr. Wu:

I'm pleased to inform you that your manuscript has been deemed suitable for publication in PLOS ONE. Congratulations! Your manuscript is now with our production department. 

Kind regards, 

on behalf of

Dr. Daniel M. Johnson 

Academic Editor

PLOS ONE